# SC83288 is a clinical development candidate for the treatment of severe malaria

Stefano Pegoraro[1], Maëlle Duffey[2,3], Thomas D. Otto[4], Yulin Wang[2,3,†], Roman Rösemann[5], Roland Baumgartner[1], Stefanie K. Fehler[1,2,3], Leonardo Lucantoni[6], Vicky M. Avery[6], Alicia Moreno-Sabater[7,8], Dominique Mazier[7,9], Henri J. Vial[10], Stefan Strobl[5], Cecilia P. Sanchez[2,3] & Michael Lanzer[2,3]

Severe malaria is a life-threatening complication of an infection with the protozoan parasite *Plasmodium falciparum*, which requires immediate treatment. Safety and efficacy concerns with currently used drugs accentuate the need for new chemotherapeutic options against severe malaria. Here we describe a medicinal chemistry program starting from amicarbalide that led to two compounds with optimized pharmacological and antiparasitic properties. SC81458 and the clinical development candidate, SC83288, are fast-acting compounds that can cure a *P. falciparum* infection in a humanized NOD/SCID mouse model system. Detailed preclinical pharmacokinetic and toxicological studies reveal no observable drawbacks. Ultra-deep sequencing of resistant parasites identifies the sarco/endoplasmic reticulum $Ca^{2+}$ transporting PfATP6 as a putative determinant of resistance to SC81458 and SC83288. Features, such as fast parasite killing, good safety margin, a potentially novel mode of action and a distinct chemotype support the clinical development of SC83288, as an intravenous application for the treatment of severe malaria.

[1] 4SC AG, Am Klopferspitz 19a, 82152 Martinsried, Germany. [2] Department of Infectious Diseases, Parasitology, Universitätsklinikum Heidelberg, Im Neuenheimer Feld 324, 69120 Heidelberg, Germany. [3] German Center for Infection Research (DZIF), partner site Heidelberg, 69120 Heidelberg, Germany. [4] Parasite Genomics, Wellcome Trust Sanger Institute, Wellcome Genome Campus, Hinxton CB10 1SA, UK. [5] 4SC Discovery GmbH, Am Klopferspitz 19a, 82152 Martinsried, Germany. [6] Eskitis Institute for Drug Discovery, Griffith University, Don Young, Nathan Queensland 4111, Australia. [7] Sorbonne Universités, UPMC Univ Paris 06, INSERM U1135, CNRS ERL 8255, Centre d'Immunologie et des Maladies Infectieuses (CIMI-Paris), 91 Bd de l'hôpital, F-75013 Paris, France. [8] AP-HP, Hôpital St Antoine, Service de Parasitologie-Mycologie, F-75012 Paris, France. [9] AP-HP, Groupe hospitalier La Pitié-Salpêtrière, Service de Parasitologie-Mycologie, F-75013 Paris, France. [10] Dynamique des Interactions Membranaires Normales et Pathologiques, CNRS UMR 5235, Université Montpellier II, cc107, Place Eugène Bataillon, 34095 Montpellier, France. † Present address: Department of Parasitology, College of Basic Medical Sciences, Dalian Medical University, 9 South Lvshun Road Western Section, Dalian, Liaoning 116044, China. Correspondence and requests for materials should be addressed to S.P. (email: stefano.pegoraro@4sc.com) or to M.L. (email: michael.lanzer@med.uni-heidelberg.de).

The decline in global malaria mortality rates by 48% over the previous decade has raised hopes of radical malaria control[1]. However, the burden of malaria is still high. There were 213 million cases of malaria and 438,000 deaths in 2015 alone, with 3.2 billion people at risk[1]. Moreover, much of the success gained against malaria has been owed to the excellent therapeutic efficacy of artemisinin derivatives and their partner drugs, which together as an artemisinin-based combination therapy (ACT) have formed the backbone of malaria control since 2005 (ref. 2). However, the clinical efficacy of ACT is under attack. *Plasmodium falciparum* strains have emerged that display a delayed and possibly reduced responsiveness to ACT[3]. Such early stage ACT resistant parasites have recently spread from their place of origin at the Thai/Cambodian border across all of Southeast Asia and run the risk of spreading further into Asia and beyond[4]. The consequences would be devastating since alternatives to ACTs are not readily available.

The evolving lack of chemotherapeutic options against malaria accentuates the need for new antimalarial drugs. Such novel drugs should exploit novel molecular targets and have distinct chemical structures to protect them from cross-resistance mechanisms. Several candidates that meet these criteria are in the development pipeline[5–14]. However, many more would be needed to compensate for the high attrition rate expected during the clinical trials and to build up a stock of reserve antimalarial drugs that can quickly replace first-line drugs should they fail.

In addition to novel drugs for the treatment of uncomplicated malaria, there is an equally urgent need for a new generation of drugs for the treatment of severe malaria. Severe malaria is a life-threatening condition that inevitably leads to the death if not treated immediately. The leading symptoms of severe malaria include impaired consciousness, vital organ dysfunction, hyperparasitaemia and the inability to take oral medicine. Severe malaria is currently treated with intravenous (i.v.) or intramuscular artesunate for at least 24 h or, if parenteral artesunate or artemether are not available, with i.v. quinine[15]. However, in addition to the resistance problem outlined above, there are concerns regarding the safety of both treatment regimens. i.v. artesunate has recently been associated with delayed haemolysis in 7–21% of the treated patients[16,17] and quinine is known to cause tinnitus and other audiovisual disorders[18].

We have recently shown that amicarbalide exerts a potent antiplasmodial activity *in vitro*, with a half-maximal inhibitory concentration ($IC_{50}$) of 10 nM (ref. 19). Amicarbalide is a benzamidine derivative that was used in veterinary medicine as an antiprotozoal drug against babesiosis, theileriosis and anaplasmosis[20,21]. Slow parasite clearance rates, high relapse frequencies, poor oral bioavailability, undesirable mutagenic and toxic side effects, and manufacturing safety issues were the reasons why this drug is no longer in use[21,22]. Here we describe a medicinal chemistry program starting from amicarbalide that identifies two agents with improved pharmacological and antiplasmodial properties. The two compounds, SC81458 and SC83288, are fast-acting trophocidal drugs that eliminate *P. falciparum* blood stage parasites both *in vitro* and in a humanized non-obese diabetic/severe combined immunodeficiency (NOD/SCID) gamma c (NSG) mouse model system. Detailed preclinical pharmacokinetic (PK) and toxicological studies support the clinical development of SC83288 towards an i.v. application for the treatment of severe malaria. We further link the $Ca^{2+}$ pump PfATP6 to reduced susceptibility to SC83288 and SC81458.

## Results

**Optimization of benzamidines as antiplasmodials.** Benzamidines are generally associated with poor oral bioavailability, which is explained by the strong basic character of the amidine group and the formation of a stable cation[23,24]. To improve oral bioavailability several strategies have been employed, including mimetic replacement of the amidine group. We initially followed this approach, starting with the East side of amicarbalide, while maintaining the West-side amidine group and the benzamidino-ureido-phenyl scaffold (Fig. 1). All compounds were tested *in vitro* for their activity against asexual blood stages of the multi-drug-resistant *P. falciparum* strain Dd2 (ref. 25).

The removal of the East-side amidino group or its complete substitution by an amine, amide or ester group abolished the antiplasmodial activity (Supplementary Table 1). We therefore explored a wider range of substituents including carbonyl-amino, amino-sulfonyl and amino-carbonyl groups, but decided to focus on 4′-sulfonamides because of their overall higher antiplasmodial activity and the ease to synthesize a large number of derivatives. A total of 20 sulfonamide derivatives were synthesized and analysed in a reiterative process. The best activity ($IC_{50}$ of $10 \pm 3$ nM, $n = 3$; mean ± s.e. of the mean of $n$ independent determinations) was obtained for compound SC09064, where a bulky 4′-sulfonamidomethyl benzene para-sulfonamido group was introduced (Fig. 1). The compound had improved physicochemical and absorption, distribution, metabolism, excretion (ADMET) properties compared with the parent molecule amicarbalide, including better solubility in water (316 versus 100 μg ml$^{-1}$), improved metabolic stability in a human microsome assay (65% residual after 1 h of incubation), negative in Ames test, and no adverse effects on HepG2 cells at a concentration of 10 μM. However, the permeability as determined in CACO 2 cells ($P_{app} = 0.28 \times 10^{-6}$ cm s$^{-1}$) and the oral bioavailability (3% in rats) remained poor, which we attributed to the presence of the remaining amidine group on the West side of the molecule.

The West-side amidine group of SC09064, however, was essential for the antiplasmodial activity and replacing it with amines or uretanic groups rendered the compound inactive (Supplementary Table 1). We therefore explored a series of modifications of the amidine group and found that functionalizing it with a piperazine ring produced compounds with high antiplasmodial activity (Fig. 2; Supplementary Table 1). However, introducing the piperazine ring did not significantly improve the permeability, falling short of the targeted value of $P_{app} > 10 \times 10^{-6}$ cm s$^{-1}$. The best two compounds in terms of activity and permeability were SC81458 and SC83288 with $IC_{50}$ values of $8 \pm 1$ nM ($n = 6$) and $3 \pm 1$ nM ($n = 6$), and $P_{app}$ values of $0.76 \times 10^{-6}$ and $0.42 \times 10^{-6}$ cm s$^{-1}$, respectively (Table 1; Fig. 3a; Supplementary Table 2). Additional criteria driving the medicinal chemistry optimization were solubility in water (Supplementary Table 2) and therapeutic efficacy in a rodent malaria model system (see below). Synthesis scheme and analytical chemistry of SC81458 and SC83288 are depicted in Supplementary Figs 1 and 2. Figure 2 summarizes the structure-activity relationship of all 172 derivatives evaluated during the course of the hit to lead optimization campaign.

**SC81458 and SC83288 are fast-acting antiplasmodial compounds.** Due to the steep dose–response curves, SC81458 and SC83288 have favourable $IC_{90}$ (18 and 8 nM, respectively) and $IC_{99}$ (50 and 20 nM, respectively) values in parasite growth assays (Fig. 3a; Table 1). SC81458 and SC83288 were also active against a range of drug-resistant *P. falciparum* lab strains other than Dd2, with $IC_{50}$ values consistently being < 20 nM (Supplementary Table 3). These data suggest that SC81458 and SC83288 are able to overcome established antimalarial drug resistance mechanisms.

Both compounds were also active against early stage (I–III) gametocytes, with $IC_{50}$ values of $76 \pm 6$ nM ($n = 3$) and

**Figure 1 | Structures and *in vitro* activities of hit and lead compounds.** The names of the compounds, their half maximal inhibitory concentrations (IC$_{50}$), and their molecular masses are indicated.

$199 \pm 30$ nM ($n = 3$), respectively; however, they showed negligible activity against late stage (IV and V) gametocytes (SC81458 IC$_{50} = 1.8 \pm 0.2$ μM, $n = 3$; SC83288 IC$_{50} > 30$ μM). The activities against liver and insect stages were not determined.

Because of their excellent *in vitro* activity against *P. falciparum* asexual blood stages, together with the opportunity to develop a needed alternative i.v. treatment for severe malaria, we further evaluate SC81458 and SC83288 despite their lack of oral bioavailability. We first investigated the stage-specific activity of the two compounds. To this end, highly synchronized cultures of Dd2-containing rings (2–4 h post invasion), trophozoites (24–26 h post invasion) and schizonts (35–37 h post invasion) were exposed to different concentrations of the compound for 6 h. Cells were subsequently washed and placed in drug-free medium until time point 10 h post invasion of the following cycle, when [$^3$H] hypoxanthine was added for 24 h to determine cell viability. Under these conditions, both SC81458 and SC83288 exerted the highest activity against trophozoites with IC$_{50}$ values of $37 \pm 3$ and $40 \pm 4$ nM, respectively ($n = 4$; Fig. 3b,c). A slightly lower activity was observed against schizonts, with IC$_{50}$ values of $70 \pm 10$ and $110 \pm 30$ nM, respectively ($n = 4$; Fig. 3a,c). The activity against ring stages was only moderate, with IC$_{50}$ values of $5 \pm 1$ and $3 \pm 1$ μM, respectively ($n = 4$; Fig. 3b,c). Note that IC$_{50}$ values have to be interpreted in the context of the exposure time—6 h in the stage-specific assays versus 72 h in the standard proliferation assays.

To assess how quickly the two compounds were able to kill the parasite, we exposed highly synchronized trophozoites (24–30 h post invasion; 0.5% parasitemia) of Dd2 to different drug concentrations ranging from 0.1 to 10 μM (ref. 7). Aliquots were withdrawn at different time points (from 0 to 6 h) after the

addition of the drug and analysed for cell viability by [$^3$H] hypoxanthine incorporation. Parasite clearance depended on both drug concentration and exposure time (Fig. 4a). For instance, SC83288 cleared a 0.5% parasitemia at a concentration of 500 nM within an exposure time of $3 \pm 1$ h ($n = 4$). Artemisinin revealed a slower killing efficacy for trophozoites under these conditions (Fig. 4a).

Since this assay underestimates parasite recrudescence, we adopted a second, standardized protocol that assessed parasite viability as a function of drug exposure time based on limiting serial dilutions of treated parasites and re-growth monitoring[26]. For comparative reasons, artemisinin and atovaquone were analysed in parallel. Each drug was applied at a fixed concentration of 10 times its IC$_{50}$ value in Dd2, which were 80 and 30 nM for SC81458 and SC83288, respectively, and 19 and 10 nM for artemisinin and atovaquone, respectively. SC81458 and SC83288 cleared 99.9% of the initial parasite population (parasite clearance time (PCT$_{99.9\%}$)) within $37 \pm 4$ h and $51 \pm 6$ h, respectively, corresponding to a parasite reduction rate of $3.4 \pm 0.4$ and $3.0 \pm 0.5$ log phases over a period of 48 h (logPRR (parasite reduction ratio); $n = 4$; Fig. 4b; Table 1). Both compounds acted quickly with a lag time of $<5$ h. In comparison, atovaquone is less potent as indicated by a PCT$_{99.9\%}$ of $89 \pm 7$ h, a logPRR of $2.1 \pm 0.4$ and a lag time of $>24$ h ($n = 4$). Artemisinin, however, displayed a higher activity in this assay with a PCT$_{99.9\%}$ of $21 \pm 3$ h, a logPRR of $4.5 \pm 2$ and a zero lag time ($n = 4$)[26].

**Safety profile of SC81458 and SC83288.** Profiling SC81458 and SC83288 against a panel of standardized cellular, biochemical and

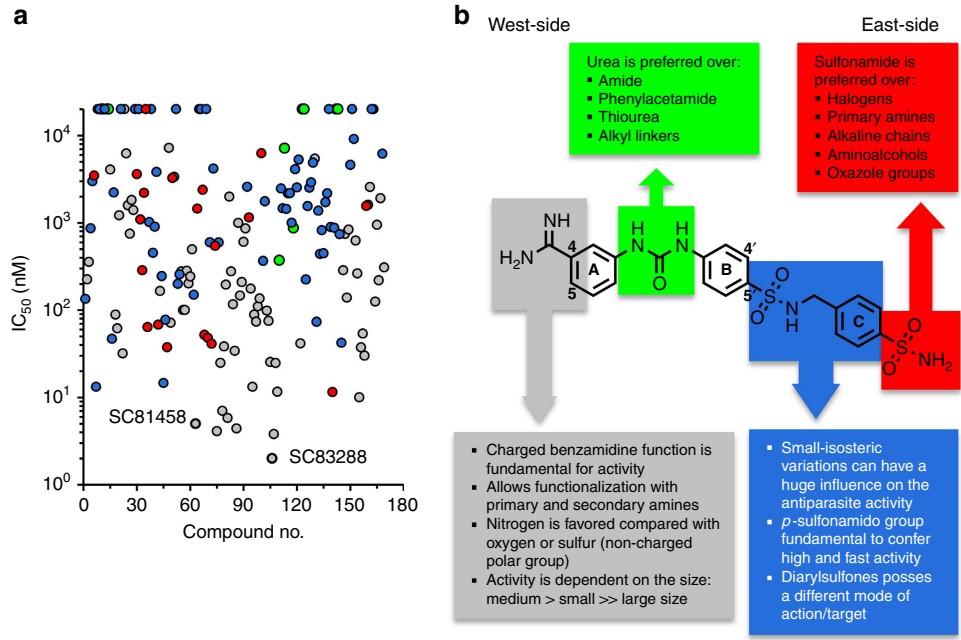

**Figure 2 | Structure-activity relationship (SAR) analysis.** (**a**) Antiplasmodial activity of 172 chronologically ordered amicarbalide derivatives against the *P. falciparum* strain Dd2. Substitutions on the West-side amidine group (grey), the benzamidine urea linker (green) and the East-side sulfonamide group (dark blue and red) are indicated. The two compounds SC81458 and SC83288 are highlighted. (**b**) Summary of the SAR analysis. Colour code as above.

| Table 1 | Relevant activity parameters of SC81458 and SC83288. | |
| --- | --- | --- |
| | **SC81458** | **SC83288** |
| $IC_{50}$, $IC_{90}$, $IC_{99}$ (nM) | 8, 18, 50 | 3, 8, 20 |
| *In vitro* $PCT_{99.9\%}$ (h) | 37 ± 4 (4) | 51 ± 6 (4) |
| *In vitro* logPRR | 3.4 ± 0.4 (4) | 3.0 ± 0.5 (4) |
| *In vitro* lag phase (h) | <5 | <5 |
| *In vivo* $PCT_{99.9\%}$ (h)* | <48 | 48 |
| *In vivo* logPRR* | 5.3 (W2), 3.2 (3D7) | 3.0 |

$IC_{50/90/99}$, inhibitory concentration of 50, 90 or 99% of the maximal effect; $PCT_{99.9\%}$, clearance time of 99.9% of the original parasite population; PRR, parasite reduction ratio.
For the *in vitro* $PCT_{99.9\%}$ and the *in vitro* logPRR the means ± s.e. of the means of (*n*) independent determinations are shown.
*Obtained in the *P. falciparum*-infected humanized NSG mouse model system.

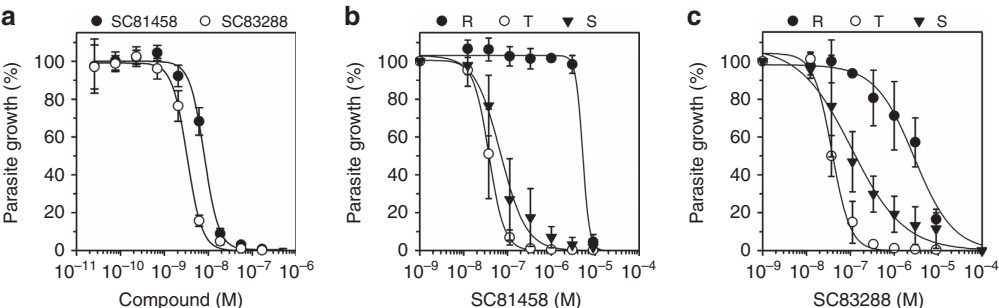

**Figure 3 | Susceptibility of the *P. falciparum* strain Dd2 to SC81458 and SC83288.** (**a**) Growth inhibition by SC81458 (closed circles) and SC83288 (open circles) in a standard cell proliferation assay with an exposure time of 72 h. (**b,c**) Growth inhibition by SC81458 (**b**) and SC83288 (**c**) against rings (R, closed circles), trophozoites (T, open circles) and schizonts (S, closed inverted triangles) after an exposure time of 6 h. The mean ± s.e.m. are shown for at least six independent replicates.

physiological assays revealed no obvious off-target activities. The two compounds did not affect *in vitro* propagation of Balb C 3T3 cells, HepG2 cells, human T-cells, or human peripheral blood macrophages or showed any other signs of cytotoxicity at a fixed concentration of 50 μM. They (10 μM) did not interfere with any of the 401 human kinases examined in a competition assay (Supplementary Table 4). The binding potential to 54 human

transporters and receptors was low to moderate, with none of the interactions occurring in the pharmacologically relevant concentration range (Supplementary Table 5). SC81458 and SC83288 did not block human cardiac hERG-mediated $K^+$ conductance. hERG retained >90% of its activity at a concentration of 60 and 20 μM for SC81458 and SC83288, respectively. At a concentration of 100 μM of SC83288 the

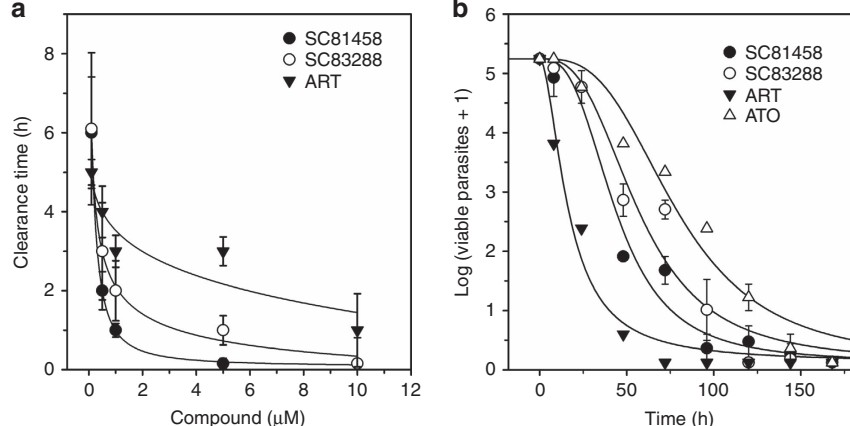

**Figure 4 | *In vitro* killing kinetic profile of SC81458 and SC83288.** (**a**) Concentration and exposure time-dependent clearance of a 0.5% parasitemia of trophozoites. The following drugs were investigated: artemisinin (ART, closed inversed triangles), SC81458 (closed circles) and SC83288 (open circles). (**b**) The graph shows the change in the number of viable parasites over time after exposure to ART (closed inversed triangles; 19 nM), atovaquone (ATO, open triangles; 10 nM), SC81458 (closed circles; 80 nM), and SC83288 (open circles; 30 nM) at a concentration corresponding to 10 times their respective $IC_{50}$ values. The mean ± s.e.m. are shown for at least four independent replicates.

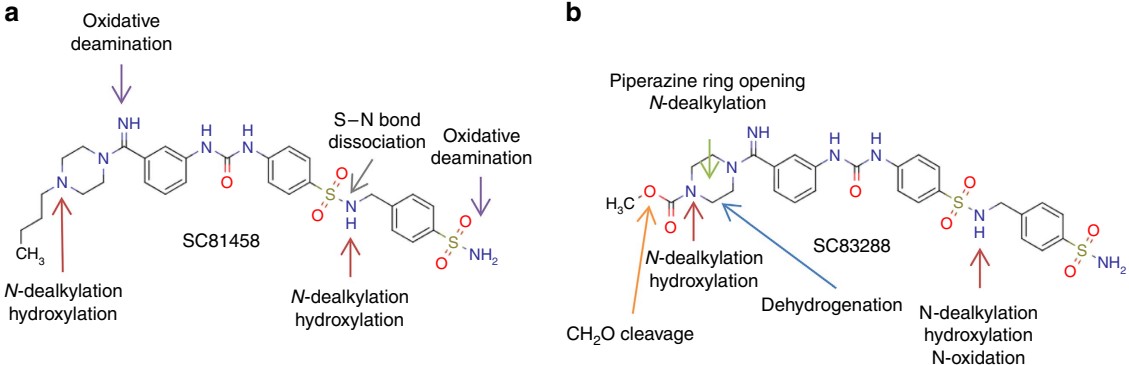

**Figure 5 | Stability of SC81458 and SC83288 in liver microsomes.** Suggested metabolic reactions and biotransformation sites of SC81458 (**a**) and SC83288 (**b**). Further detail is provided in Supplementary Figs 5 and 6.

channel remained 50% active. The inhibition potential of SC81458 and SC83288 (10 μM) towards 13 human cytochrome P450 enzymes was low and only the CYP2E1 isoform was modestly inhibited by 30% by SC83288 (Supplementary Table 6). SC81458 and SC83288 showed no signs of genotoxicity, at 50 μM, in a reverse Ames test or of mutagenic or clastogenic activity in an *in vitro* micronucleus assay at 60 μM.

The maximal-tolerated i.v. bolus dose of SC83288 was at 30 mg kg$^{-1}$ body weight (extrapolated initial drug plasma concentration $C_0$ of 65,600 ng ml$^{-1}$) and 15 mg kg$^{-1}$ body weight ($C_0$ of 27,300 ng ml$^{-1}$) in rats and mice, respectively. Observed adverse effects included ataxia, respiratory symptoms and sleepiness. However, all treated animals fully recovered within 30 min after drug application. Haematological and clinical biochemical parameters assessed 24 h post application remained within normal range in all animals (Supplementary Table 7). Increasing the dose to 45 mg kg$^{-1}$ body weight caused apnoea in rats. The no-observed-adverse-effect level and the no-observed-effect level in rats corresponded to a tested i.v. bolus dose of 22.5 mg kg$^{-1}$ body weight ($C_0$ of 49,200 ng ml$^{-1}$) and 15 mg kg$^{-1}$ body weight ($C_0$ of 32,800 ng ml$^{-1}$), respectively.

**Stability and *in vitro* metabolism.** SC81458 and SC83288 were soluble in water (890 and 940 μg ml$^{-1}$, respectively) and organic solvents. Measured pK values were 5.72 and 9.78 for SC81458 and 9.82 for SC83288. The solubility in phosphate buffer decreased

with increasing pH, from 1.8 mM at pH 4 to 0.4 mM at pH 9 for SC81458 and from 2.6 mM at pH 4 to 0.8 mM at pH 9 for SC83288 (Supplementary Fig. 3). The compounds were stable in different physiological solutions, including artificial gastric juice, simulated intestinal fluid, phosphate-buffered saline and human plasma, when incubated for 6 h at 37 °C.

The *in vitro* elimination and metabolic profile of SC81458 and SC83288 were determined using mouse, rat, dog, monkey and human liver microsomal incubations with initial compound concentrations of 10 μM. SC81458 was almost completely metabolized by monkey and human liver microsomal extracts during the 60 min incubation time, with an apparent intrinsic clearance of 303 μl min$^{-1}$ mg$^{-1}$ (Supplementary Fig. 4a). The compound was more stable in dog, mouse and rat microsomal extracts, with 37%, 54% and 74% of the initial amount remaining after 60 min of incubation, respectively (Supplementary Fig. 4a). In comparison, SC83288 was metabolically more stable. The remaining abundances of SC83288 in mouse and rat liver microsomal extracts were 96% and 91%, respectively (Supplementary Fig. 4b). In dog, monkey and human liver microsomal extracts, 83%, 51% and 50% of the compound remained after the 60 min incubation period, respectively (Supplementary Fig. 4b). In human microsomal incubations, SC83288 declined with an apparent intrinsic clearance of 75.8 ± 2.5 μl min$^{-1}$ mg$^{-1}$ ($n=3$). The disappearance of SC81458 and SC83288 was mainly cofactor dependent and thus

driven by metabolic activity in all species (Supplementary Fig. 4a,b).

The metabolite profiles of SC81458 and SC83288 were similar in all species, with minor differences in terms of the relative distribution of metabolites and the overall number of metabolites generated (Fig. 5; Supplementary Figs 5 and 6). The main biotransformations were hydroxylations (or $N$-oxidations), dehydrogenations, and $N$-dealkylations and more specifically, the cleavage of the $N$-butyl piperazine chain and the cleavage of the 'East-side' 4-sulfamoylphenyl methylene group. The resulting two major metabolites were fully characterized for their antiplasmodial activity *in vitro*. Metabolite 1 (M6 and M10 for SC81458 and SC83288, respectively; see Supplementary Figs 5 and 6) showed an $IC_{50}$ value of 65 nM and metabolite 2 (M7 and M18 for SC81458 and SC83288, respectively) of 5.0 μM. Both major metabolites did not reveal off-targets effects in the standard assays described above.

SC81458 and SC83288 were stable in the presence of cultured human red blood cells and in *P. falciparum* cultures for at least 6 h, as indicated by the absence of metabolites or break-down products, suggesting that neither the human erythrocytes nor the parasite metabolically alter the compound (Supplementary Fig. 7).

**SC81458 and SC83288 cure a *P. falciparum* infection in mice.** We next evaluated the efficacy of SC81458 and SC83288 in humanized NSG mice infected with the wild-type *P. falciparum* strain 3D7 or the multi-drug-resistant strain W2. At day 7 post infection when all mice had developed a patent parasitemia, mice were treated intraperitoneally (i.p.) with 0, 2.5, 5.0 and 10.0 mg kg$^{-1}$ body weight of SC81458 or SC83288 once per day over the next 4 days. In all treatment groups, parasitemia dropped rapidly without an apparent lag phase and fell below detectable levels between day 8 and 10 post infection (with the exception of 3D7-infected mice treated with 2.5 mg kg$^{-1}$ per day of SC81458; Fig. 6). Maximal kill rates were observed at 10.0 mg kg$^{-1}$ per day i.p., with SC81458 and SC83288 reducing the parasite burden by ∼97 and 93% within the first 24 h. Over a replicative cycle of 48 h, SC83288 reduced the parasitemia by 99.9% or three log phases (*in vivo* logPRR of 3.0 and *in vivo* PCT$_{99.9\%}$ of 48 h) with no distinction in activity between the two parasite strains tested (Table 1; Supplementary Fig. 8a). In comparison, SC81458 was more effective against W2 than 3D7 and the *in vivo* logPRR ranged from 5.3 to 3.2 for the two strains (Table 1). All mice were surveyed for possible parasitemia for the next 3 weeks. However, no recrudescence of the infection occurred, with the exception of 3D7-infected mice treated with 2.5 mg kg$^{-1}$ of SC81458. None of the animals showed signs of discomfort or intolerability during and after the treatment regimens.

The two SC compounds were also active in the *Plasmodium vinckei* rodent malaria model system. SC83288 administered i.p. at a dose of 20 mg kg$^{-1}$ once per day for four consecutive days fully cured the infection (initial parasitemia of 1.0% at time of first administration). All treated mice survived the initial infection and there was no recrudescence. In the case of SC81458, a dosing regimen of 30 mg kg$^{-1}$ per day for 4 days reduced the parasite burden by 97% (89.9% and 3.3% parasitemia in untreated and treated mice at day 5, respectively). However, the infection persisted and all treated mice died of a recrudescent parasitemia in the days following the last administration of the compound.

Contrasting with the efficacy against *P. falciparum* and *P. vinckei*, SC81458 and SC83288 were inactive against the rodent malaria parasite *Plasmodium berghei*. This was the case for both an *in vivo* 4 day dosing regimen of 30 mg kg$^{-1}$ per day and an *ex vivo* one-cycle growth inhibition assay. Similarly, the parental compound SC09064 (Fig. 1) revealed no activity against *P. berghei* and it was also inactive against *P. vinckei* (see Supplementary Table 8 for results on all derivatives tested in rodent malaria model systems). Differential therapeutic efficacy in various rodent model systems is a well-established phenomenon and is typically associated with charged compounds, such as positively charged amidines and negatively charged phosphoric acid or carboxylic acid containing drugs[27–29]. The phenomenon is explained based on selective uptake of charged compounds through parasite-induced channels, termed new permeation pathways, in the host cell plasma membrane and species-specific characteristics of the new permeation pathways[27,28]. Consistent with these previous findings, derivatives containing a free and hence protonated amidine group, such as SC09064, were generally inactive in the rodent malaria model systems, whereas compounds in which the amidine was functionalized as a piperazine revealed therapeutic efficacy against *P. vinckei* but not against *P. berghei in vivo* (Supplementary Table 8).

**PKs of SC81458 and SC83288.** Figure 7a,b depict the PKs of SC81458 and SC83288 after a single i.p. dose in mice. In the case of SC83288, the same dosages as used in the efficacy study were administered, namely 2.5, 5.0 and 10.0 mg kg$^{-1}$. For SC81458, a dose of 20 mg kg$^{-1}$ was investigated. In all cases, the plasma concentrations ($C_{max}$) peaked within 30 min or less after drug application, suggesting a rapid entrance of the drug into the systemic circulation. The plasma concentrations then rapidly declined with an effective half time $t_{1/2}$ of 52 min for SC81458 and 29 to 55 min for SC83288 (Table 2). The $AUC_{0-inf}$ (area under the curve) were 4,342 and 2,019 ng h$^{-1}$ ml$^{-1}$ for the highest doses of SC81458 and SC83288 examined.

To better understand the PK/pharmacodynamic (PK/PD) relationship of SC83288, we analysed the *in vivo* growth inhibition data (taken from Fig. 6d, time point: 2 days after commencement of treatment) as a function of the free plasma concentration, the latter being calculated from the $C_{max}$ values by correcting for 89% plasma protein binding of SC83288 in mice (Supplementary Table 9). A sigmoidal half-maximal effective dose ($ED_{50}$) model was fitted to the resulting data points, yielding estimates for the $ED_{50}$ and $ED_{90}$ of 45 and 80 nM, respectively (Supplementary Fig. 8b). The free-minimal parasiticidal concentration (MPC) at which parasite growth was completely inhibited was 200 nM for SC83288, in good agreement with the *in vitro* growth inhibition data (Supplementary Fig. 8a, see Fig. 2a for comparison).

A detailed i.v. PK of SC83288 in four different species revealed that therapeutic plasma exposures can be achieved after a bolus i.v. application (2 mg kg$^{-1}$ body weight in mice, rats and cynomolgus monkeys and 1.7 mg kg$^{-1}$ body weight in dogs; Fig. 7c). For instance, in cynomolgus, SC83288 reached a $C_0$ of 7,338 ng ml$^{-1}$, which corresponds to ∼17 times the free MPC. The plasma protein binding of SC83288 was 82% in cynomolgus (Supplementary Table 9). The plasma level declined with a $t_{1/2}$ $_{initial}$ of 16 min, a $t_{1/2}$ $_{terminal}$ of 320 min and an effective $t_{1/2}$ of 62 min. SC83288 was cleared from the blood of the non-human primate with a rate of 470 ml h$^{-1}$ kg$^{-1}$. None of the animals showed any signs of intolerance during or after i.v. application of SC83288 at the concentrations indicated above. The PK properties of SC83288 were distinct from those of the parental compound SC09064 and the close amicarbalide relative imidocarb with regard to area under the curve, plasma half-life, clearance and volume distribution (Supplementary Fig. 9).

On the basis of the PK data in the four animal species, we predicted the total plasma clearance of SC83288 in humans by allometric scaling, taking into account the maximal life potential

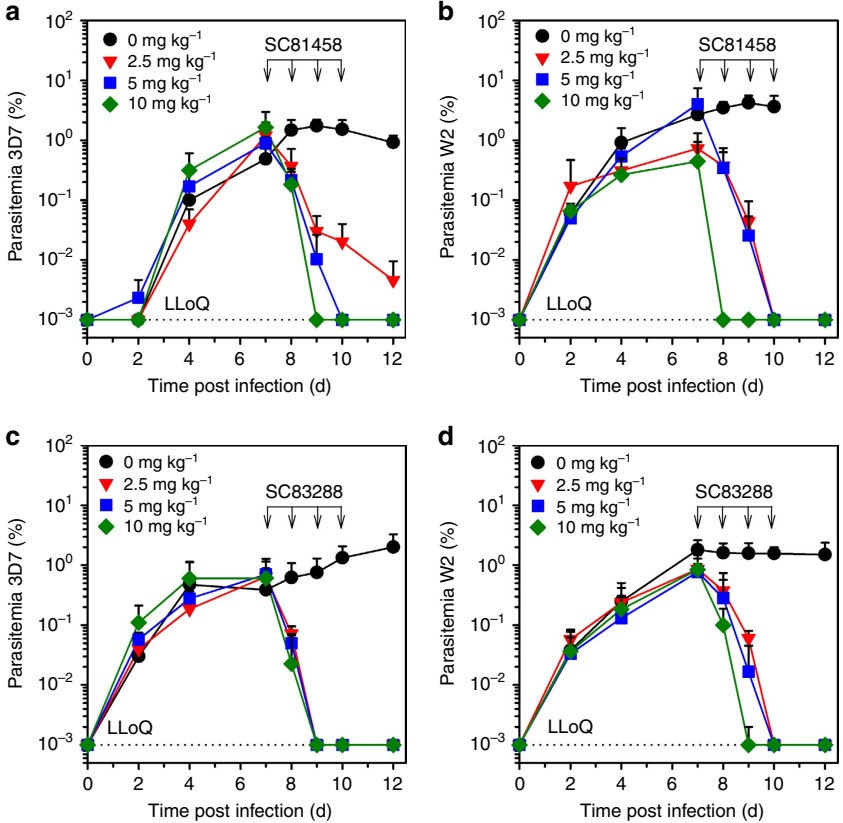

**Figure 6 | In vivo efficacy of SC81458 and SC83288 in *P. falciparum*-infected humanized NSG mice.** NSG mice engrafted with human erythrocytes were infected with the *P. falciparum* strains 3D7 (**a**,**c**) or W2 (**b**,**d**). The compounds SC81458 (**a**,**b**) or SC83288 (**c**,**d**) were administered i.p. once per day over 4 days starting on day 7 post infection. Parasitemia was assessed each day from day 2 post infection up to 12 days. The mean ± s.e.m. of four mice in each treatment group are shown.

of the respective species and by applying the empirical approach of the rule of exponents[30]. This yielded a predicted human plasma clearance of $14.7 \, l \, h^{-1}$ for SC83288 (Fig. 7d). We further fit a minimal physiologically based PK (mPBPK) model to the plasma data of the preclinical species to predict the human PK profile and to simulate different infusion schemes[31] (Supplementary Fig. 10). The mPBPK model yielded a total body clearance of $15.2 \, l \, h^{-1}$ ($13.1 \, l \, h^{-1}$ with plasma binding correction). Both an i.v. bolus injection of 25 mg or an infusion of $15 \, mg \, h^{-1}$ can result in therapeutically relevant plasma exposures, according to the model (Supplementary Fig. 10).

**Resistance to SC81458 and SC83288.** To gain insights into the mode of action and the mechanism of resistance, we attempted to select mutant lines. No resistant parasites emerged during exposure of $10^8$ asexual blood stages to 100 nM compound for 60 days. These data suggest that the frequency of spontaneous resistance emergence is very low, possibly below the single-point mutation rate of $10^{-9}$. However, exposing a starting number of $10^{10}$ asexual blood stages to gradually increasing drug concentrations (from 50 nM to 1,000 nM) selected for resistant parasite lines during a period of ∼240 days (Fig. 8a,b). Eight clonal parasite lines displaying different levels of resistance were obtained by limiting dilution and subjected to ultra-deep sequencing (Fig. 8a,b; Supplementary Table 10). All clones displayed cross resistance to both SC compounds, suggesting that SC81458 and SC83288 share a common mode of action and/or the parasites possess the same mechanism of resistance (Fig. 8c). Resistance to the SC compounds did not decrease susceptibility to licensed antimalarial drugs (Supplementary Table 11).

Of the polymorphisms and copy number variations detected in the drug selected clones three correlated with decreased susceptibility to SC81458 and SC83288 (Fig. 8d). These polymorphisms were coding mutations in *Pf3D7_1447900* encoding the multi-drug resistance transporter 2 (PfMDR2; four out of eight clones with mutation), *Pf3D7_0106300* encoding the $Ca^{2+}$ transporting PfATP6 (all clones had different gene modifications, six clones had mutations and four clones had duplications of the gene) and *Pf3D7_1241800* encoding a putative ATP-dependent RNA helicase DBP9 (six clones; Supplementary Fig. 11).

ATP-dependent RNA helicases DBP9 are generally involved in RNA metabolism, including ribozyme assembly, splicing and translation initiation[32]. The non-synonymous mutation identified occurred in a repetitive region, changing a glutamic acid to aspartic acid at position 533 (Supplementary Fig. 11a).

PfMDR2 belongs to the ABC transporter superfamily and is localized at the parasite's plasma membrane and, possibly, the digestive vacuolar membrane[33,34]. PfMDR2 confers heavy metal resistance and it contributes to decreased susceptibility to atovaquone, mefloquine and quinine but not chloroquine[35,36]. PfMDR2 is dispensable during asexual replication[36]. The non-synonymous mutation found in several SC81458 and SC83288 selected clones results in the substitution of the positively charged amino-acid lysine for the negatively charged amino-acid glutamic acid at position 412 in transmembrane domain 6 (Supplementary Fig. 11b). It is tempting to speculate that the introduction of the negative charge allows PfMDR2 to act on the positively charged SC compounds and expel them from the cell.

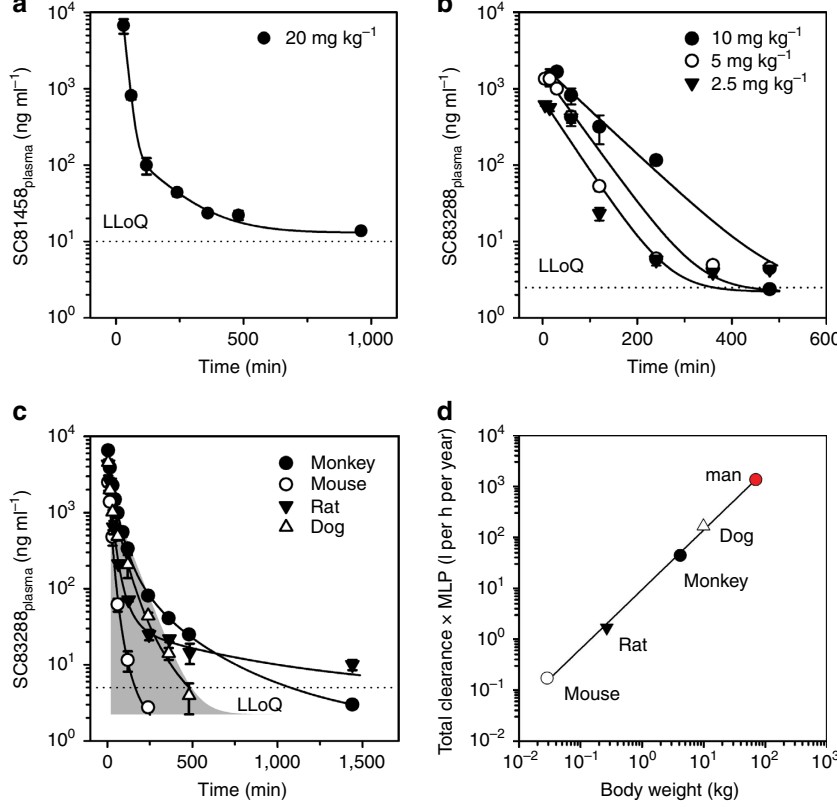

**Figure 7 | Pharmacokinetic profiles of SC81458 and SC83288. (a,b)** Mean plasma concentration of SC81458 (**a**) and SC83288 (**b**) over time following i.p. administration of the indicated doses in mice. (**c**) Mean plasma concentration of SC83288 over time following i.v. administration of $2\,mg\,kg^{-1}$ in monkeys (closed circles), mice (open circles), and rats (closed inverted triangles) and of $1.7\,mg\,kg^{-1}$ in dogs (open triangles). For comparative reasons, the SC83288 AUC of the i.p. $10\,mg\,kg^{-1}$ treatment group in mice is indicated in grey. The dotted lines show the lower limit of quantification (LLoQ). The mean $\pm$ s.e.m. are shown for at least three animals. Data were examined using a non-compartment analysis (NCA) and relevant PK parameters are compiled in Table 2. (**d**) Four species allometric scaling of the plasma clearance rate (CL) of SC83288, according to Ring et al.[30]. The CL values were corrected for the maximum life potential (MLP). The function $y = ax^b$ was fitted to the data points, yielding values for $a$ of 9.47 and $b$ of 1.17 ($R^2 = 0.997$). Extrapolation of the fit provided estimates of the human blood clearance rate (red circle).

**Table 2 | Pharmacokinetic parameters of SC81458 and SC83288.**

| Compound | SC81458 | | | | SC83288 | | | |
|---|---|---|---|---|---|---|---|---|
| Application | i.p. | | | | i.v. | | | |
| Species | Mouse | | | | Mouse | Rat | Dog | Cynomolgus |
| Dose ($mg\,kg^{-1}$) | 20 | 2.5 | 5 | 10 | 2 | 2 | 1.7 | 2 |
| $AUC_{0-inf}$ ($ng\,h\,ml^{-1}$) | 4,342 | 646 | 1,089 | 2,019 | 891 | 1,419 | 2,262 | 4,253 |
| $C_0$ ($ng\,ml^{-1}$) | 0 | 0 | 0 | 0 | 3,644 | 4,375 | 5,579 | 7,338 |
| $C_{max}$ ($ng\,ml^{-1}$) | 7,384 | 610 | 1,338 | 1,711 | 2,560 | 3,283 | 4,564 | 6,569 |
| $t_{max}$ (min) | 30 | 5 | 5 | 30 | 5 | 5 | 5 | 5 |
| $t_{1/2\ initial}$ (min) | NC | NC | NC | NC | 9 | 9 | 12 | 16 |
| $t_{1/2\ terminal}$ (min) | 860 | 100 | 73 | 46 | 43 | 625 | 90 | 320 |
| $t_{1/2\ effective}$ (min) | 52 | 33 | 29 | 55 | 12 | 52 | 35 | 62 |
| $MRT_{0-t}$ (h) | 1.24 | 0.79 | 0.69 | 1.33 | 0.28 | 1.25 | 0.84 | 1.50 |
| CL ($ml\,h^{-1}\,kg^{-1}$) | 4,607 | 3,870 | 4,590 | 4,953 | 2245 | 1,410 | 884 | 470 |
| $V_C$ ($ml\,kg^{-1}$) | NC | NC | NC | NC | 549 | 457 | 359 | 273 |
| $F$ (%) | 62 | 58 | 49 | 45 | 100 | 100 | 100 | 100 |

AUC, area under the concentration-time curve; $C_0$, initial drug plasma concentration; $C_{max}$, observed maximum plasma concentration after administration; CL, total plasma clearance of drug after administration; F, relative bioavailability compared to i.v.; MRT, mean residence time of the unchanged drug in the systemic circulation; NC, not calculated; $t_{max}$, time of $C_{max}$; $t_{1/2}$, time required for the concentration to fall to 50% of its initial value; $V_C$, apparent volume of the central or plasma compartment.
Data were examined using a non-compartment analysis.

PfATP6 plays a crucial role in $Ca^{2+}$ homoeostasis in *P. falciparum* by acting as the sarco/endoplasmic reticulum (ER) $Ca^{2+}$ pump and by facilitating $Ca^{2+}$ transport across the parasite's digestive vacuolar membrane. The polymorphisms identified include alanine to threonine substitutions at positions 108 and 109 within transmembrane domain 2 for SC83288 selected clones and, for SC81458 selected clones, a substitution of tyrosine for phenylalanine at position 972 within the large cytoplasmic loop (Supplementary Fig. 11c). In addition to point mutations, the PfATP6 locus and flanking regions on

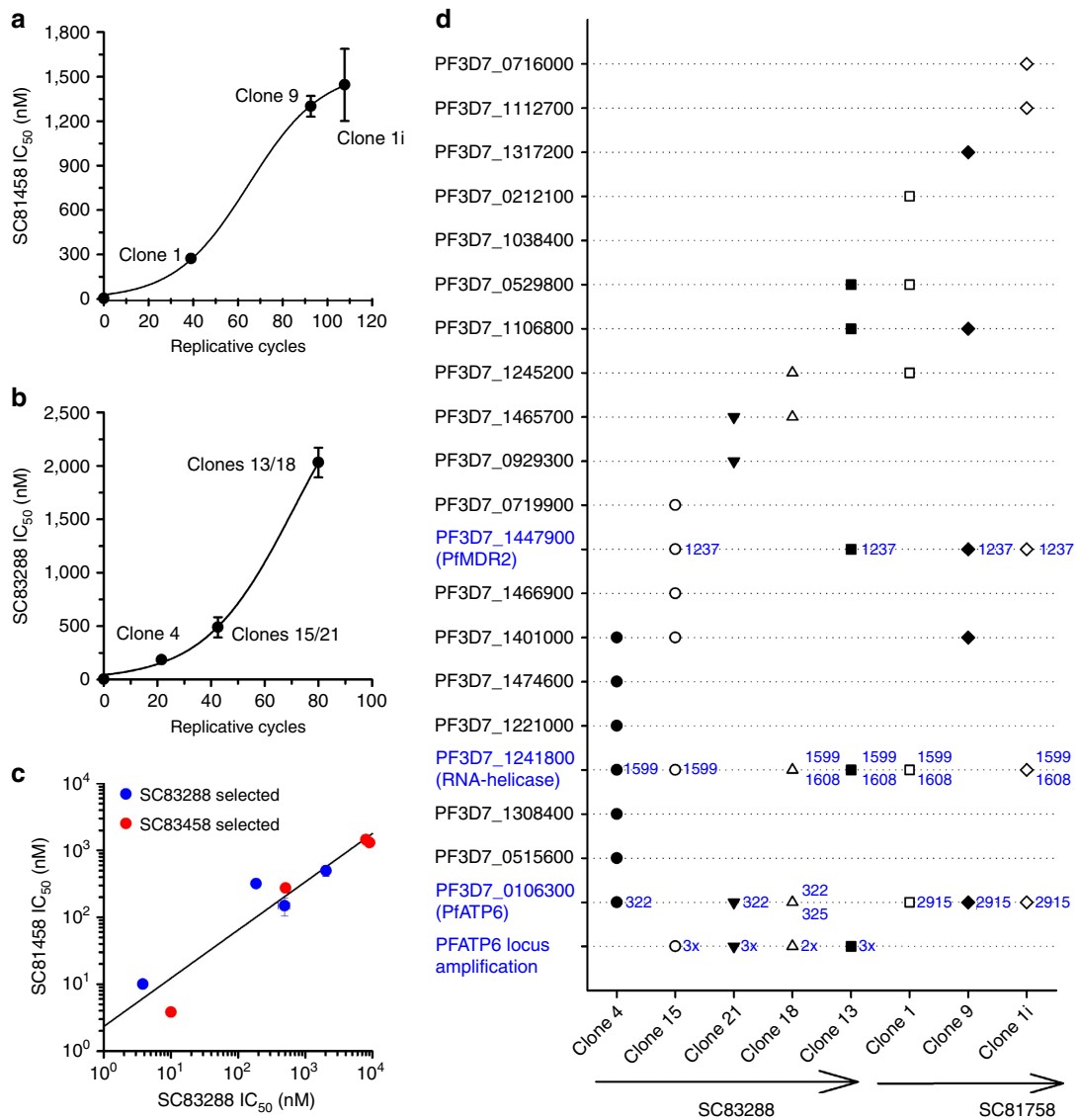

**Figure 8 | In vitro generation of SC81458 and SC83288-resistant P. falciparum clones.** (**a,b**) $\sim 10^{10}$ infected erythrocytes were treated with increasing concentration of SC81458 (**a**) or SC83288 (**b**) and the decrease in drug susceptibility was monitored over time. Clones obtained at different time points during selection are indicated. The means ± s.e.m. of at least three independent determinations are shown. (**c**) Cross resistance between SC81458- and SC83288-selected clones. (**d**) Identified single-nucleotide polymorphisms (SNPs) and gene amplifications in the genome of resistant clones. The arrows point in the direction of increased selective pressure. SNPs and gene amplifications associated with resistance to SC compounds are highlighted in blue. This includes PfMDR2 (position 1237), a putative ATP-dependent RNA helicase DBP9 (position 533) and PfATP6 (position 322 and 325), as well as an amplification of the PfATP6 locus.

chromosome 1 were duplicated in SC83288 selected clones (Fig. 8d). The duplications were of different nature. One clone had the complete chromosome 1 duplicated, two clones shared the same domain duplicated, but all four clones duplicated a total of six genes, including PfATP6 (Supplementary Fig. 11d).

To understand if any of the mutations identified in the three candidate genes might exist already in the *P. falciparum* population, we checked the 3,500 samples from the MalariaGen consortium (https://www.malariagen.net/data/catalogue-genetic-variation-p-falciparum-v4.0). However, none of the mutations were found; all were *de novo*.

The presence of both point mutations and gene amplification is striking and suggests a critical role of PfATP6 in SC83288 resistance and, possibly, the antiplasmodial mode of action. If SC83288 and SC81458 target PfATP6 then inhibition of this pump would incapacitate the ER of its function as a $Ca^{2+}$ store

and the concentration of free $Ca^{2+}$ would increase in the parasite's cytoplasm. To investigate this hypothesis, we loaded *P. falciparum*-infected erythrocytes with the $Ca^{2+}$ sensitive fluorochrome Fluo-4 and recorded dynamic changes in cytoplasmic free $Ca^{2+}$ in a live cell confocal set-up. However, treatment of the cells with SC83288 (10 and 20 μM) or SC81458 (10 and 20 μM) did not affect cytoplasmic free $Ca^{2+}$ (Supplementary Fig. 12). In comparison, cyclopiazonic acid (CPA; 10 μM), an established inhibitor of PfATP6 (ref. 37), induced a strong $Ca^{2+}$ responses in the parasite's cytoplasm (Supplementary Fig. 12). These findings suggest that PfATP6 plays a role in resistance to the SC compounds, but it does not seem to be the molecular target.

## Discussion

SC81458 and SC83288 fulfil several criteria of the target candidate profile for a molecule against severe malaria[38]. Both compounds

have advantageous parasitological properties. They display steep, in vitro grow inhibition curves with $IC_{50}$ values $< 10$ nM and $IC_{99}$ values $\leq 50$ nM (Table 1) and absolute Hill coefficients larger than unity ($2.6 \pm 0.3$ for both compounds, Fig. 2a). We interpret this concentration–response behaviour as evidence of a biological threshold phenomenon. SC81458 and SC83288 act quickly, killing $\geq 99.9\%$ of the parasites within the first 48 h of treatment as demonstrated in a standardized in vitro assay[26,38] and in the humanized NSG mouse model system (Figs 4 and 6). The parasite reduction rate of three log units is a conservative estimate given that, in human patients, parasite killing is aided by the host's immune system and splenic entrapment and clearance mechanisms[26]. A logPRR of $\geq 3.0$ is considered a favourable property of an antimalarial drug candidate[38], although the value is lower than that of artemisinin[26]. While the compounds are primarily active against replicative blood stages, with trophozoites being the most susceptible stage, they also target early stage (I–III) gametocytes. Being devoid of activity against the late stage gametocytes (IV and V) the compounds may not exert appreciable transmission blocking activity. In vitro preclinical profiling discovered no obvious liabilities. Moreover, all animals tolerated exposure to the compounds at therapeutic concentrations without any signs of distress or discomfort. However, a full evaluation of the safety of the compounds awaits further toxicological studies, in particular acute and 28 day repeat-dose toxicology in a non-rodent species.

Both SC81458 and SC83288 belong to a chemotype that has not yet been used in malaria chemotherapy and that does not reveal cross resistance to currently used antimalarials.

The analysis of SC83288- and SC81458-resistant clones provided first clues about the possible resistance mechanisms and/or the mode of action. Among the three candidate genes associated with resistance Pf3D7_0106300 encoding the $Ca^{2+}$-transporting PfATP6 stands out. The finding of both point mutations and gene duplications suggests that Pf3D7_0106300 is under strong selective pressure by the SC compounds. PfATP6 is the sarco/ER $Ca^{2+}$ pump of the parasite, and as such plays a vital role in $Ca^{2+}$ homoeostasis. PfATP6 was once considered a putative target of artemisinin[39,40]. However, more recent studies failed to validate this hypothesis[41] and, instead, point towards a mode of action of artemisinin that involves inhibition of a phosphatidylinositol-3-kinase[42].

SC81458 and SC83288 do not seem to target PfATP6. At least no $Ca^{2+}$ responses were induced in the parasite's cytoplasm on the addition of the SC compounds, as would have been expected if SC81458 and SC83288 were to directly inhibit PfATP6. This suggests a putative role of PfATP6 in the drug resistance mechanism, possibly by acting as a compensatory mechanism for the yet to be identified molecular target of the SC compounds. Alternatively, mutated and amplified PfATP6 might affect the intracellular distribution of the compounds by transporting the compounds into a compartment, where the compounds are less harmful to the parasite.

A comparable role in resistance can be envisaged for PfMDR2, which when mutated might expel the SC compounds from the cell. A function of PfMDR2 as an export system is consistent with the subcellular localization of PfMDR2 at the parasite's plasma membrane[33,34] and the fact that PfMDR2 confers heavy metal tolerance and contributes to decreased susceptibility to several antimalarial drugs including quinine and atovaquone[35,36].

The role of the putative ATP-dependent RNA helicase DBP9 in the mode of action/mechanism of resistance of the SC compounds is less clear (Fig. 8). RNA helicases have not yet been considered drug targets in malaria research, although specific inhibitors have been developed in other systems[43].

Although both SC compounds are able to cure a P. falciparum infection in humanized NSG mice, SC83288 showed improved metabolic stability, better efficacy over a wider dose range and less strain variability, compared with SC81458 (Fig. 6; Table 1)—therefore nominating SC83288 as a potential preclinical development candidate. The dose finding experiments conducted in the mouse challenge model, together with the resulting PK/PD relationship, provide first estimates of dose projections of SC83288 in humans. Taking into consideration that the initial treatment against severe malaria is i.v. applied until the patient is able to take oral medication for final parasite elimination, the mouse PK/PD data suggest that a steady-state free plasma concentration of $1.5 \times$ the free MPC of 200 nM maintained over a period of $< 8$ h suffices to reduce the initial parasite burden by 99.9% within 48 h (Supplementary Fig. 8a). Assuming a parasitemia of 10% that is not uncommon in patients with severe malaria, the total parasite burden will fall from its initial value of $\sim 10^{12}$ infected erythrocytes (assuming a total red blood cell count of $2–3 \times 10^{13}$) to $\sim 10^{9}$ or below if the additional contribution of splenic clearance and immune killing mechanisms are considered. The parasite burden may even fall faster with longer exposure times, as suggested by the in vitro killing speed experiments revealing concentration and exposure-dependent parasite killing kinetics (Fig. 4b). A free plasma concentration of $1.5 \times$ the free MPC of 200 nM translates to a total plasma concentration of 860 ng ml$^{-1}$ (assuming 78% plasma protein binding, Supplementary Table 9). Maintaining such a plasma concentration would require infusion rates between 8.8 and 13.1 mg h$^{-1}$ given a total plasma clearance rate between 1,0250 ml h$^{-1}$ (lower estimate) and 15,180 ml h$^{-1}$ (higher estimate; Fig. 7d). How long such an infusion should be administered or whether i.v. or intramuscular bolus applications suffice would await efficacy studies in clinical settings. The projected dosing regimen would maintain a safety margin of $\sim 55$-fold assuming comparable i.v. no-observed-adverse-effect level in rats and humans with a $C_0$ of 49,200 ng ml$^{-1}$. In summary, SC83288 may not match the favourable pharmacological and PDs properties of artemisinin in all aspects. However, the attributes of SC83288 have to be considered in light of the emerging resistance against artemisinin derivatives[3,4] and the adverse side effects observed in a substantial number of patients treated with parenteral artesunate[16,17]. The role of SC83288 as an alternative chemotherapeutic option will require extensive clinical studies. However, the available preclinical data suggest that SC83288 holds promise for an efficacious and safe clinical candidate for the acute treatment of severe malaria.

## Methods

**Ethical clearance.** All animal studies were carried out in strict accordance with national and European guides for the care and use of laboratory animals (European regulations; 2010/63/EU). The challenge studies using humanized NSG mice were approved by the Centre d'Expérimentation Fonctionnelle (CEF, La Pitié-Salpêtrière, Paris) and the Ministère de l'Education Nationale, de l'Enseignement Supérieur et de la Recherche (authorization number 01737.03). All procedures pertaining to the PK studies were in compliance with the German Animal Welfare Act and German regulations (TierSchG/TierSchVersV) and were approved by the Regierung von Oberbayern, München, (authorization number 55.2-1-54-2532.2-9-11; studies in mice and rats) and the Niedersächsisches Landesamt für Verbraucherschutz und Lebensmittelsicherheit, Oldenburg (authorization numbers: 33.2-42502-05-LG-01-86/2012 and 33.2-42502-05-LG-01/2014/04 SA; studies in cynomolgus monkeys and dogs).

**Chemicals.** The following chemicals were purchased from the following vendors: Sigma-Aldrich: sodium/potassium ATPase inhibitor ouabain, artemisinin, atovaquone, 4-(chlorosulphonyl)phenyl isocyanate, 4-nitro-sulphonyl-4-aniline and acetonitrile. Fisher Scientific: isopropanol, dimethylacetamide, dichloromethane, dimethylformamide, diisopropylethylamine, 3-nitrobenzamidine, 3-aminobenzamidine; tin(II) dichloride dihydrate, formic acid, acetonitrile, 4-trifluoromethylaniline, 3-chloroaniline, 2-bromoaniline, 2-aminobenzonitrile,

2-bromo-4,6-difluoroaniline, 3-tri-fluoromethyl-4-chloraniline, 3-trifluoromethylaniline, 2-bromo-4-trifluoromethylaniline, 3,6-bis-trifluoro-methylaniline, 2,4-di-bromoaniline, butylamine, benzyl-amine, adamantylamine, oxalamide and sulphamide. Acros Organics: 4-aminosulfonamide, 3-isocyanatobenzonitrile, n-butylpiperazine, 4-hydroxypiperidine, piperazine, silica gel, 4-(4-bromophenylsulfonyl)aniline, 4-(4-nitrophenylsulfonyl)aniline, 4-(4-chlorophenylsulfonyl)aniline and methyl sulfoxide-d6. Maybridge Chemical Comp. Ltd.: 1,1-dioxo-1H-benzothiophen-6-ylamine, 4-benzene-sulphonylpheny-lamine and 3H-benzimidazol-5-ylamine. Molecular Probes: Fluo-4-AM, Pluronic F-127 and Probenecid. The sarco/ER calcium ATPase inhibitor CPA was purchased from Calbiochem.

**Synthetic scheme for SC81458 and SC83288.** SC81458 and SC83288 were part of a focus library consisting of 172 compounds. Compounds were synthesized in groups of 6 to 12 in small-scale laboratory preparations. Each compound was purified by liquid chromatography, in most cases by preparative high-performance liquid chromatography/mass spectrometry (HPLC/MS). Before being screened *in vitro* each compound identity was confirmed by analytical HPLC/MS. If the compound was active against *P. falciparum* blood stages compound identity was confirmed by [1]H nuclear magnetic resonance (NMR; Supplementary Note 1). In addition, an [13]C-NMR spectrum was collected for a number of key compounds, including the hit compound SC09064 and the two lead compounds SC81458 and SC83288 (see below and Supplementary Note 1). Analytical HPLC–MS determinations were performed with Waters 2,700 Autosampler, Waters 1,525 Multisolvent Delivery System and Micromass ZQ single quadrupole mass spectrometer with electrospray source. Column: Chromolith Fast Gradient C18 (Merck), $50 \times 2$ mm, with stainless steel $2 \mu$m prefilter. Eluent A, $H_2O + 0.1\%$ HCOOH; eluent B, MeCN. Preparative HPLC–MS were performed with a Waters 2,700 Autosampler, Waters 600 Multisolvent Delivery System with preparative pump heads ($500 \mu$l), Waters 600S Controller and Waters ZQ single quadrupole mass spectrometer with electrospray source. Column: Waters X-Terra RP18, $5 \mu$m, $19 \times 150$ mm. Eluent A, $H_2O + 0.1\%$ HCOOH; eluent B, MeCN. Different linear gradients, individually adapted to sample. Bioanalytical HPLC–MS for PK analysis were performed with Agilent 1100 HTLC System, HTS-Pal Autosampler and Micromass Ultima triple quadrupole, mass spectrometer with electrospray source. Column: Chromolith RP18, $50 \times 4.6$ mm. Eluent A, $H_2O + 0.1\%$ HCOOH; eluent B, MeCN. [1]H-NMR and [13]C-NMR were performed with a Bruker AV300 (300.13 Mhz) at a temperature of 305 K. The compounds described in this document were prepared according to methods described in the following synthesis method. Steps (i and ii): production of the cyanourea compound (4). An amount of 28 mmol of 3-aminobenzonitrile (1) were dissolved in 40 ml dichloromethane (DCM). An amount of 27 mmol of 4-isocyanatobenzene-1-sulfonyl chloride (2) were added portionwise over 20 min and the mixture was allowed to stir at room temperature (RT) overnight. The urea product (3) precipitated, was filtered, washed with cold DCM and was used directly in the next step (yield 93%). One equivalent of 4-(aminomethyl)benzene-1-sulfonamide hydrochloride was dissolved in MeCN and 1.5 equivalent N,N-diisopropylethylamine (DIEA) was added. The solution was cooled to 0 °C and, under vigorous stirring, 1 equivalent of the ureido-sulfonylchloride compound was added portionwise. The reaction was allowed to stir overnight, allowing the temperature to increase to RT. The solvent was eva-porated and the product (4) was obtained by precipitation with ethylacetate/petrol ether or MeOH/diethylether. If necessary, the product was further purified by flash chromatography using a gradient of MeOH in DCM mixtures as eluent. $[M + H]^+$ calculated for $C_{21}H_{19}N_5O_5S_2$, 486.090; found for $[M + H]^+$: 486.1. [1]H-NMR (300 MHz, DMSO-$d_6$) $\delta = 3.97$ (d, $J = 6.2$ Hz, 2 H), 7.22 (bs, 2H), 7.36–7.47 (m, 3H), 7.57–7.69 (m, 7H), 7.91 (t, $J = 1.7$ Hz, 1H), 8.01 (t, $J = 6.3$ Hz, 1H), 9.06 (s, 1H), 9.19 (s, 1H). [13]C-NMR (75 MHz, DMSO-$d_6$) $\delta = 45.54$, 111.64, 118.00 ($2 \times$), 118.74, 121.09, 123.17, 125.56 ($2 \times$), 125.71, 127.81 ($4 \times$), 130.20, 133.23, 140.19, 141.98, 142.91, 143.08, 152.16. Steps (iii and iv): conversion of the cyano com-pound to functionalized benzamidine. Under argon atmosphere, 1 equivalent of the cyanoureido (4) compound was dissolved with dry MeOH. An amount of 4 M HCl in dioxane was added and the solution was stirred overnight at RT. The solvent was removed; the residue was taken in diethylether which was evaporated. This operation was repeated three times to yield either a solid or an oily product. A measure of 1 equivalent of the dried benzimidate was dissolved in N,N-dime-thylformamide (DMF) and variable amounts of the chosen secondary amine were added (1.0–1.1 equivalents). The solution was stirred overnight at temperatures ranging from 55 to 70 °C. The solvent was removed and the product was purified by preparative HPLC and a gradient of acetonitrile in 0.1% HCOOH aq.

SC81458 (4-(3-(3-(((4-butylpiperazin-1-yl)(imino)methyl)phenyl)ureido)-N-(4-sulfamoylbenzyl) benzenesulfonamide) was prepared from 4-(3-(3-cyanophenyl)ureido)-N-(4-sulfamoylbenzyl) benzenesulfonamide using 1.1 equivalents of n-butylpiperazine as secondary amine. The product was purified by preparative HPLC. Yield: 21%; $[M + H]^+$ calculated for $C_{29}H_{37}N_7O_5S_2$, 628.237; found for $[M + H]^+$: 628.1; [1]H-NMR (300 MHz, DMSO-$d_6$) $\delta = 0.88$ (t, $J = 7.2$ Hz, 3H), 1.24–1.33 (m, 2H), 1.35–1.46 (m, 2H), 2.32–2.36 (m, 2H), 2.50 (m, 4H under DMSO peak), 3.52–3.60 (bm, 4H), 4.03 (bs, 2H), 7.13 (d, $J = 7.6$ Hz, 1H), 7.27–7.36 (m, 2H), 7.44 (d, $J = 8.3$ Hz, 2H), 7.50 (t, $J = 7.9$ Hz, 1H), 7.69–7.76 (m, 8H), 7.83–7.86 (m, 1H), 7.99–8.12 (m, 1H), 8.53 (s, 1H), 11.19–11.29 (m, 2H). [13]C-NMR (75 MHz, DMSO-$d_6$) $\delta = 13.81$, 19.95, 28.23, 45.56 ($3 \times$), 51.76 ($2 \times$),

56.78, 116.99, 117.57 ($2 \times$), 120.75, 121.18, 125.55 ($2 \times$), 127.66 ($2 \times$), 127.81 ($2 \times$), 129.64, 129.85, 132.42, 141.09, 142.02, 142.91, 144.17, 152.94, 163.94, 167.24 ($HCOO^-$). element. anal.: element (calc. comp. incl. add. (%)/found (%)): C: (53.56/51.82), H: (5.69/5.9), N: (14.57/14.16), S: (9.53/9.2). Oxygen not determined.

SC83288 (methyl 4-(imino(3-(3-(4-(N-(4-sulfamoylbenzyl)sulfamoyl)phenyl) ureido) phenyl)methyl) piperazine-1-carboxylate) was prepared from 4-(3-(3-cyanophenyl)ureido)-N-(4-sulfamoylbenzyl)benzenesulfonamide using 1 equivalent of methyl piperazine-1-carboxylate as secondary amine. The product was purified by preparative HPLC. Yield: 13%; $[M + H]^+$ calculated for $C_{27}H_{31}N_7O_7S_2$, 630.180 ; found for $[M + H]^+$: 630.1; [1]H-NMR (300 MHz, DMSO-$d_6$) $\delta = 3.28$–3.32 (m, 2 H), 3.55–3.62 (m, 6H), 3.64 (s, 3H), 4.03 (d, $J = 6.1$ Hz, 2H), 7.21 (d, $J = 7.5$ Hz, 1H), 7.29 (bs, 2H), 7.44 (d, $J = 8.3$ Hz, 2H), 7.54 (t, $J = 7.9$ Hz, 1H), 7.63–7.78 (m, 7H), 7.86–7.89 (m, 1H), 8.08 (t, $J = 6.3$ Hz, 1H), 9.39 (bs, 1H), 9.90 (s, 1H), 9.96 (s, 1H). [13]C-NMR (75 MHz, DMSO-$d_6$) $\delta = 42.23$ ($2 \times$), 45.55 ($3 \times$), 52.58, 117.43, 117.74 ($2 \times$), 121.46, 121.60, 125.56 ($2 \times$), 127.80 ($4 \times$), 129.63, 129.81, 132.97, 140.37, 142.01, 142.91, 143.47, 152.41, 154.92, 164.39. element. anal.: element (calc. comp. incl. add. (%)/found (%)): C: (48.68/48.25), Cl: (5.32/4.77), H: (4.84/5.6), N: (14.72/14.51), O: (16.81/18.1), S: (9.63/7.99).

**Permeability assay.** The study was conducted by Cyprotex (Macclesfield, United Kingdom). CACO 2 cells (Sigma-Aldrich, catalogue: 86010202) were seeded on Millipore Millicell plates and formed a confluent monolayer over 20 days before the experiment. On day 20, the tested compounds at a final concentration of $10 \mu$M were added to the apical side of the membrane. The transport of the compound across the monolayer was monitored over a 2 h period by liquid chromatography (LC)–MS/MS quantification. The permeability coefficient ($P$app) was calculated from the following equation: $P_{app} = \frac{dQ/dt}{C_0 \times A}$, where $dQ/dt$ is the rate of permeation of the drug across the cells, $C_0$ the donor compartment concentration at time zero and $A$ the area of the cell monolayer. $C_0$ was obtained from analysis of the dosing solution at the start of the experiment.

**Cell culture.** *P. falciparum* parasites were maintained in continuous *in vitro* culture as previously described[44]. Briefly, blood cultures were grown in 10 ml petri dishes at 37 °C under controlled atmospheric conditions of 3% $CO_2$, 5% $O_2$, 92% $N_2$ and at a humidity of 95%. Cells were grown at a haematocrit of 5.0% and at a parasitemia of no >5%.

**IC$_{50}$ measurement.** Growth inhibition assays were performed according to standard protocols based on the detection of parasitic DNA by fluorescent SYBR green staining[45] or by [[3]H] hypoxanthine incorporation[25]. Briefly, a previously sorbitol-synchronized culture of *P. falciparum* Dd2 ring-stage parasites was incubated in the presence of decreasing drug concentrations in a 96-well black microtiter plate, at the following conditions: 100 μl per well, 0.5% parasitemia, 2% haematocrit, incubation at 37 °C. If parasite proliferation was assayed using SYBR green, plates were incubated for 72 h, before being frozen at − 80 °C overnight. On the day of the measurement, plates were thawed for 1 h at RT. An amount of 100 μl of a 1 × SYBRGreen (ThermoFisher Scientific Inc.) solution in lysis buffer was added to each well. The plates were then briefly shaken and incubated at RT for 1 h in the dark. Fluorescence was measured in a fluorescence plate reader (FluoStar Optima, BMG Labtech GmbH; ext/em: 485/520 nm, gain 1,380, 10 flashes per well, top optic). If parasite proliferation was assayed using [[3]H] hypoxanthine incorporation, 0.5 μCi [[3]H]-hypoxanthine was added to each well over the last 24 of the 72 h incubation period. Cells were collected on glass-filter plates, which was dried at 60 °C for 1 h. An amount of 20 μl of scintillation cocktail was added and the incorporated radioactivity was determined using a microtiter plate scintillation counter. The IC$_{50}$ values were calculated using SigmaPlot 13 (Systat Software Inc.) according to the Hill function (four parameters).

**In vitro speed of action.** The *in vitro* speed of action of SC81458, SC83288, artemisinin and atovaquone on Dd2 *P. falciparum* parasites was assessed according to two different protocols: first, we adapted the previously established protocol by Sanz *et al.*[26] and performed the assay using standardized conditions. Briefly, asynchronous cultures (0.5% parasitemia and 2% haematocrit) with a predominant ring population were treated with the selected drugs at concentrations corresponding to 10 times their respective IC$_{50}$ values. Drugs were renewed daily over the entire treatment period. Samples of untreated (0 h) and treated parasites (8, 24, 48, 72, 96, 120, 144 and 168 h) were aliquoted to perform serial dilutions (dilution factor of 3, 12 dilution points) in 96-well plates in triplicates by adding fresh erythrocytes and fresh culture medium. Before the serial dilution the drugs were washed out using fresh culture medium. Parasites were cultured for 21 days to allow wells with viable parasites to render detectable parasitemia, observed by SYBRGreen staining as described above. The number of viable parasites in the original aliquot was back calculated by using the formula $X^{n-1}$, where $n$ is the number of wells able to render growth and $X$ is the dilution factor. The logarithm of viable parasites +1 was plotted against time and fitted to a four parameter

logistic curve using SigmaPlot 13 (Systat Software Inc.). From there, the lag phase, the logPRR, and the $PCT_{99.9\%}$ were determined for each drug.

The second approach followed the protocol described by Vial et al.[7]. Briefly, highly synchronized trophozoite-stage parasites (24–26 h post invasion; 0.5% parasitemia, 2% haemtocrit) were treated with 100, 500, 1, 5 and 10 μM of artemisinin, SC81458 and SC83288 for up to 6 h (0 h, 10 min, 30 min, 1 h, 2 h, 3 h, 4 h, 5 h and 6 h), before the drugs were washed out and cells returned to culture. At time point 10 h post invasion of the following cycle, [³H]-hypoxanthine was added for 24 h to monitor parasite viability[46]. The clearance time was determined for each treatment condition and was expressed as a function of the drug concentration.

### In vitro metabolism in liver microsomes.

The study was conducted by Novamass Ltd (Oulu, Finland). The in vitro disappearance and metabolite profile of SC81458 and SC83288 was determined using mouse, rat, dog, monkey and human liver microsomal incubations with initial substrate concentrations of 10 μM. Briefly, SC81458 and SC83288 were incubated with liver microsomes (0.5 mg of microsomal protein per millilitre) in the presence of the appropriate cofactor. Two parallel incubates, one with cofactor and one without, were employed. Each reaction mixture was preincubated for 2 min at 37 °C. Reaction was started by addition of 1 mM NADPH and 1 mM UDPGA. After incubation period of 0 or 60 min, samples were collected and the reaction was terminated by addition of ice-cold acetonitrile. Samples were subsequently cooled in an ice bath and centrifuged (10 min, 16,100g). The supernatants were transferred to maximum recovery vials and analysed by LC/MSMS.

### Determination of efficacy in the Pf/NSG mouse model.

NSG male mice aged 9–11 weeks (Charles River, US) were used for this study. They were kept in sterile isolators, provided with UV light-exposed commercial food and autoclaved water ad libitum. Human red blood cells (HRBC) were obtained from the Etablissement Français du Sang (Ile-de-France, Rungis). HRBC were washed twice with RPMI-1640 medium (Gibco/BRL) by centrifugation at 900g, 10 min at 25 °C resuspended in RPMI-1640 medium and kept at 4 °C for a maximum of 2 weeks.

Mice humanization and P. falciparum infection was carried out as previously described[47]. Mouse tissue macrophages were depleted by clodronate, provided by Roche Diagnostics (Mannheim,Germany), encapsulated in liposomes (lip-clod) as previously described[48]. Neutrophils were controlled using the monoclonal antibody (mAb) NIMP-R14 (the monoclonal antibody NIMP-R14 was purified from a hybridoma provided by Dr M. Strath from the National Institute for Medical Research, London, UK)[49]. On day 0, each mouse was injected i.p. with a dose of 10 mg kg$^{-1}$ of mAb NIMPR14, mixed with 60 μl of lip-clod (300 μg of clodronate). On D2 and D4 and D6 each mouse received, i.p., 0.5 ml of HRBC mixed with a dose of 10 mg kg$^{-1}$ of mAb NIMP-R14 and 60 μl of lip-clod. On D8, each mouse was injected with 0.5 ml of HRBC infected with P. falciparum strains at a parasitemia of 0.3%, mixed with a dose of 10 mg kg$^{-1}$ of mAb NIMP-R14 and 60 μl of lip-clod. Following the P. falciparum infection, a dose of 10 mg kg$^{-1}$ of mAb NIMP-R14 and 30 μl of lip-clod was injected every 2–3 days. The haematocrit and graft of HRBC in the peripheral blood of mice was followed up during the assay in blood samples taken from the tail. After the P. falciparum infection, the HRBC graft was carried out every 2–3 days except in those mice where their haematocrit was up to 60% and the percentage of HRBC higher than 60%. Those mice only received the immunosupressor treatment and the graft of HRBC was carried out again once the haematocrit decreased to 50%.

### PK studies.

On the treatment day, all animals were weighed and the dosing volume was calculated for each individual animal according to its actual body weight. All animal were dosed once. Blood samples were taken at different time points and plasma was collected after centrifugation (10,000g, 10 min, 4 °C). Plasma samples were analysed by LC–MS/MS. During and after the application, animals were observed for clinical signs and mortality. All relevant observations were recorded. The PK studies in cynomolgus monkeys and beagle dogs were performed by the Laboratory of Pharmacology and Toxicology (LPT), Hamburg, and PK and toxicity studies in mice and rats were performed by the 4SC Discovery GmbH, Munich. PK parameters were obtained by a non-compartment analysis, using Kinetica 4.1 (Thermo Scientific).

### In vitro generation of drug-resistant P. falciparum lines.

To identify the mode of action/mechanism of resistance of SC81458 and SC82388, a chemogenomic approach was used, whereby resistant parasites were obtained by in vitro drug pressure selection and subjected to whole-genome sequencing to identify the genetic basis of resistance. Three independent selections of Dd2 parasites at an initial population of $10^{10}$ infected red blood cells (performed in 150 ml flasks, 2% haematocrit) were subjected to SC81458 or SC83288 at a starting concentration of 50 nM. The drug pressure was gradually increased by increments of 50 nM up to 1 μM according to the parasite fitness. Drug resistance was confirmed by measuring the $IC_{50}$s, after two consecutive cycles in absence of the drug, using a SYBRGreen growth inhibition assay as previously described[45]. After up to 200 days of drug pressure, the drug-resistant lines were clones by limiting dilution for single-cell isolation and genomic DNA was extracted using the DNeasy Blood&Tissue kit (Qiagen, USA) and subjected to ultra-deep sequencing.

### Whole-genome sequencing of isolates.

Genomic DNA was extracted from eight clones selected for reduced susceptibility to SC83288 and SC81458 resistant clones and the Dd2 parent. Illumina libraries of 300–500 bp fragment length were generated using a PCR-free protocol[50] and sequenced using an Illumina MiSeq obtaining 150 bp reads. The sequence data have been deposited in the European Nucleotide Archive (ENA) under the accession code ERP005793 (https://www.ebi.ac.uk/ena/data/search?query=ERP005793). The accession numbers of the reads are compiled in Supplementary Table 10.

### Bioinformatics analysis.

The analysis was performed as described[13,14]. In short, to improve mapability the P. falciparum reference genome 3D7 was transformed into the Dd2 parent clone by repeatedly mapping the reads and correcting the differences using iCORN2 (ref. 51). Next, this new reference was annotated with RATT[52]. Reads from all samples (Supplementary Table 10) were mapped against the new reference genomes using bowtie2 (ref. 53). Variants were called using two methods: (i) the genome analysis toolkit (GATK) single-nucleotide polymorphism calling pipeline[54], using the settings for Plasmodium; and (ii) mpileup and varfilter (varFilter -D 2000; Quality ≥60) from the SAMtools package[55]. A bespoke PERL script returns genes where just drug-resistant sample had single-nucleotide polymorphisms. Last, large deletions, insertions and duplications were manually detected using BAMview[56].

### Ca²⁺ live imaging.

Trophozoite-stage P. falciparum HB3 parasites were incubated with 10 μM of Fluo-4-AM in RPMI 1640 medium (Life Technologies) with Pluoronic F-127 (0.1% (v/v)) and 40 μM of probenecid for 40 min at 37 °C, as previously described[57]. Dye-loaded parasites were washed two times before settling onto poly-L-lysine-coated glass slides. Confocal laser scanning fluorescence microscopy was performed using a Leica TCS SP5 (Leica Microsystems CMS GmbH). Fluo-4-AM was excited at 488 nm (argon laser, 0.03%) and the emitted fluorescence was collected from 505–520 nm. Single images were obtained using a 63 × objective, with a fivefold software zoom, 1,024 × 1,024 pixels, every 2 s over a time span of 120 s. The Ca²⁺ ATPase inhibitor CPA, the Na⁺/K⁺ ATPase inhibitor oubain, SC81458 and SC83288 were added to a final concentration of 10 μM at 45 s. The calcium signal was further analysed using Fiji, and the mean fluorescence signals were compared using one-way analysis of variance in SigmaPlot 13 (Systat Software Inc.).

### Gametocytocidal activity.

Activity against early- and late-stage gametocytes was determined in miniaturized 384-wells plate assays as previously described[58,59]. Briefly, gametocytes from a recombinant line NF54$^{Pfs16}$, expressing a green fluorescent protein–luciferase reporter gene under the gametocyte-specific Pfs16 promoter were induced and purified as previously described[60]; gametocytes were then treated either on day 1 (stage I gametocytes) or day 8 of gametocytogenesis (stage IV gametocytes) with serially diluted SC81458 or SC83288 to achieve a final top concentration of 40 μM (0.4% dimethylsulphoxide(DMSO)). After 72 h of incubation, gametocyte inhibition was assessed by luminometry, after addition of Steadylite plus luciferase detection kit (PerkinElmer) to the plate wells[58,59].

### Plasma protein binding of SC83288.

The study was carried out by Novamass Ltd (Oulu, Finland). The plasma protein binding of SC83288 was determined in human rat, mouse and cynomolgus plasma, using a method based on rapid equilibrium dialysis and LC/MS/MS analysis. Briefly, plasma from the different species was incubated with a final concentration of SC83288 of 10 μM for 4 h at 37 °C in a rapid equilibrium dialysis device. Proteins were then precipitated by acetonitrile addition containing a phenacetin internal standard. After centrifugation (10,000g) the supernatants were analysed by LC/MS/MS to obtain the protein unbound fraction of the analyte substance.

### Inhibition of cytochrome P450 (CYP) enzymes.

The study was conducted by Novamass Ltd (Oulu, Finland) according to a previously established protocol[61]. The inhibition potential of SC81458 and SC83288 towards drug-metabolizing cytochrome P450 (CYP) was investigated using a cocktail incubation with 10 CYP-specific substrates and 13 probe reactions for nine major drug-metabolizing CYP enzymes in incubation with a pool of human liver microsomes. Briefly, each incubation mixture contained 0.5 mg microsomal protein per ml, 0.1 M phosphate buffer (pH 7.4) and the 10 probe substrates, at the following concentrations: melatonin (CYP1A2, 5 μM), coumarin (CYP2A6, 2 μM), bupropion (CYP2B6, 2 μM), amodiaquine (CYP2C8, 5 μM), tolbutamide (CYP2C9, 8 μM), omeprazole (CYP2C19 and CYP3A4, 5 μM), dextromethorphan (CYP2D6, 1 μM), chlorzoxazone (CYP2E1, 10 μM), midazolam (CYP3A4, 1 μM) and testosterone (CYP3A4, 5 μM). SC81458 and SC83288 were dissolved into DMSO and added into the incubation mixture to a final concentration of 10 μM. The reaction mixture was preincubated for 2 min at 37 °C before the reaction was initiated by addition of 1 mM NADPH. Each reaction was terminated after 20 min by addition of ice-cold acetonitrile containing a phenacetin internal standard. The samples were subsequently cooled in an ice bath to precipitate the proteins centrifuged (10 min, 16,200g). The supernatants were transferred to a Waters Max Recovery vial and analysed by LC/MSMS.

**Screening for kinase inhibition of SC81458.** The KINOMEscan was carried out by Ambit (San Diego, USA) according to a previously established protocol[62]. The experiment was based on a competition binding assay that quantitatively measures the ability of a compound to compete with an immobilized, active-site directed ligand. The assay is performed by combining three components: DNA-tagged kinase; immobilized ligand; and a test compound. The ability of the test compound to compete with the immobilized ligand is measured via quantitative PCR of the DNA tag.

For most assays, kinase-tagged T7 phage strains were grown in parallel in 24-well blocks in an *Escherichia coli* host derived from the BL21 strain. *E. coli* were grown to log-phase and infected with T7 phage from a frozen stock and incubated with shaking at 32 °C until lysis (90–150 min). The lysates were centrifuged (6,000g) and filtered (0.2 µm) to remove cell debris. The remaining kinases were produced in HEK-293 cells (Sigma Aldrich, catalogue number: 85120602) and subsequently tagged with DNA for quantitative PCR detection. Streptavidin-coated magnetic beads were treated with biotinylated small molecule ligands for 30 min at RT to generate affinity resins for kinase assays. The liganded beads were blocked with excess biotin and washed with blocking buffer (SeaBlock (Pierce), 1% bovine serum albumin, 0.05% Tween 20 and 1 mM dithiothreitol) to remove unbound ligand and to reduce non-specific phage binding. Binding reactions were assembled by combining kinases, liganded affinity beads, and SC81458 in $1 \times$ binding buffer (20% SeaBlock, $0.17 \times$ PBS, 0.05% Tween 20, 6 mM dithiothreitol). SC81458 was prepared as 40x stock in 100% DMSO and directly diluted into the assay to a final concentration of 10 µM. All reactions were performed in polypropylene 384-well plates in a final volume of 0.04 ml. The assay plates were incubated at RT with shaking for 1 h and the affinity beads were washed with wash buffer ($1 \times$ PBS and 0.05% Tween 20). The beads were then resuspended in elution buffer ($1 \times$ PBS, 0.05% Tween 20 and 0.5 µM non-biotinylated affinity ligand) and incubated at RT with shaking for 30 min. The kinase concentration in the eluates was measured by quantitative PCR.

**In vitro safety pharmacology profiling.** The study was carried out by Cerep (Celle l'Evescault, France). An *in vitro* pharmacological profiling panel was designed to detect potential high-risk clinical adverse drug reactions. The panel comprised of 54 targets. For assessment of activity against these targets, SC81458 and SC83288 were tested in an eight-point concentration–response covering up to a test concentration of 10 µM and an $IC_{50}$ and Ki were determined using radioligand binding assay.

The specific ligand binding to the receptors was defined as the difference between the total binding and the non-specific binding determined in the presence of an excess of unlabelled ligand. Results were expressed as a per cent of control specific binding and as a per cent inhibition of control specific obtained in the presence of the test compounds. Results showing an inhibition (or stimulation) >50% were considered to represent significant effects of the test compounds. 50% was the cut-off value for further investigation (determination of $IC_{50}$ or $EC_{50}$ values from concentration–response curves). The $IC_{50}$ values were determined by non-linear regression analysis of the competition curves generated with mean replicate values using Hill equation curve fitting. The inhibition constants ($K_i$) were calculated using the Cheng Prusoff equation.

**Stability in cultured blood cell cultures.** An enriched young trophozoite-stage culture of *P. falciparum* Dd2 was obtain by magnetic purification and allowed to recover for 1 h at 37 °C in RPMI 1640 medium (Life Technologies) enriched with 1% hypoxanthine (C.C.Pro GmbH) and 10% Albumax (Life Technologies). The experiment was performed in a total volume of 300 µl, at a haematocrit of 50%, in presence or absence of 2 µM of SC81458 or SC83288, for 6 h at 37 °C under shaking conditions. Each sample was subsequently treated with 1.2 ml of ice-cold acetonitrile, incubated on ice for 10 min and centrifuged for 15 min at 6,000g at 4 °C. Supernatants were stored at −80 °C until mass spectrometric measurement.

**hERG profiling.** A HEK 293 cell line (Sigma Aldrich, catalogue number: 85120602) constitutively expressing the hERG channel in the membrane was used for the assay. The cells were cultured in MEM with Earle's salts and L-glutamine (10% FCS, non-essential amino acids, 300 µg ml$^{-1}$ Geneticin (G418)) and kept at 37 °C and under 5% $CO_2$. Cells were routinely split every 3–4 days at a 1:5–1:10 ratio. The electrophysiological measurements were performed using the whole cell patch-clamp method. Extracellular bath solution contained 137 mM NaCl, 4 mM KCl, 1 mM $MgCl_2$, 1.8 mM $CaCl_2$, 10 mM glucose, 10 mM HEPES (pH 7.4 with NaOH), the intracellular pipette solution 130 mM KCl, 1 mM $MgCl_2$, 5 mM MgATP, 5 mM EGTA and 10 mM HEPES (pH 7.2 with KOH). To assess the effects of SC81458 and SC83288 on the current through the hERG channel, a 1,500 ms conditioning voltage pulse was applied every 20 s by the holding membrane potential (−80 mV) is carried out to +40 mV followed by a 500 ms test pulse of +40 mV to −40 mV. The amplitude of the current peak induced by the test pulse was determined at equilibrium ($I_0$) and after ($I_{drug}$) addition of the substance. After the application of the substance, the cells were rinsed with extracellular bath solution until a complete recovery of the current occurred; otherwise, the cells were replaced with new ones.

**Allometric scaling and prediction of human PK.** An empirical four species simple allometry according to the rule of exponents was used in predicting the human total plasma clearance of SC83288 (ref. 30). The average body weights were as following: mouse, 0.0287 kg; rat, 0.267 kg; monkey, 4.17 kg; and dog, 9.9 kg. Values for the maximum lifespan potential were taken from Boxenbaum (1982) and are: mouse, 2.66 years; rat, 4.48 years; monkey, 22.46 years; and dog, 22.20 years[63]. For man, a body weight of 70 kg and a maximum lifespan potential of 92.22 years were assumed. For a simple allometry with plasma protein binding correction only mouse, rat and monkey were used since data on plasma protein binding of SC83288 were only available for these three species. The preclinical plasma concentration versus time profiles were analysed using a mPBPK model[31]. A two tissue compartment version of the mPBPK model provided the best fits. Total plasma clearance was parameterized as $CL = aTBW^b$ (TBW, total body weight). Parameters a and b were derived from the mPBPK model. Volumes of distribution were implemented as $V1 + V2 + V_p = TBW$, where $V_p$ equals the plasma volume of each species and $V1$ was parameterized as $V1 = cTBW$. Fractions of cardiac output fd1 and fd2 satisfied the condition $fd1 + fd2 \leq 1$. Values for cardiac output were taken from Gabrielsson and Weiner[64]. A hematocrite of 0.42 was assumed for conversion to plasma flows. The parameters $a$, $b$, $c$, fd1, fd2 and $K_p$ (tissue/plasma partition coefficient) were estimated from a global fit of all data, assuming a common fd1, fd2 and $K_p$ for all species. Data below the lower limit of quantification were excluded from the fit. A $1/y^2_{pred}$ weighting scheme was applied. Assessment of model quality included visual inspection of the fitted curves, residuals sum of squares, Akaike information criterion and coefficients of variation of the estimated parameters. In the final model dog data were excluded since the data could not be fitted together with the other three species, which was possibly due to the different *in vitro* metabolite profile (Supplementary Fig. 5b). Parameters for the other species could be derived with acceptable precision (Supplementary Fig. 10b).

The mPBPK model was used to simulate the human PK and various dosing schedules to obtain anticipated pharmacological active exposures, using the i.v. route and different infusion protocols. The rate of infusion $C_{in}$ is defined by the equation $C_{in} = C_{ss} \times CL$, where $C_{ss}$ is the steady-state plasma concentration and CL is the total plasma clearance rate. The simulations were based on point estimates of the relevant parameter. Variability was not accounted for. Model fitting and simulations were performed using Phoenix64/WinNonLin 6.4 on an Intel Core i5 processor.

**Data availability.** The authors declare that the data supporting the findings of this study are available within the article and its supplementary information files, or available from the authors upon request. The whole genome sequences of the *P. falciparum* clone Dd2 and 8 *P. falciparum* clones resistant to SC83288 and SC81458 have been deposited in the European Nucleotide Archive (ENA) under the accession code ERP005793 (https://www.ebi.ac.uk/ena/data/search?query=ERP005793).

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

## Acknowledgements

The research leading to these results has received funding from the Bundesministerium für Bildung und Forschung under the portfolio of the German Centre for Infectious Research (DZIF), the Wellcome Trust (grant number 098051), the Australian Research Council (grant number LP120200557), and the European Commission under the portfolio of the FP6-integrated project AntiMal (contract number LSHP-CT-2005-0188) and the 7th framework Center of Excellence EVIMalaR. We thank Matt Berriman and Mandy Sanders (both from the Welcome Trust Sanger Institute) for processing the samples for sequencing. We thank Marina Müller and Stefan Prior (both from Heidelberg University), and Christian Portaluppi, Liliana Gustini, Sandra Rath and Sabrina Wittlinger (all from 4SC AG) for excellent technical assistance. We are grateful to Lisa Heitmann and Thomas Hesterkamp (both from the German Center for Infection Research, DZIF) for excellent project management.

## Author contributions

S.P., S.S., C.P.S. and M.L. designed the study. S.P. designed and synthesized chemical compounds. S.F. contributed to the analytical analysis of compounds. M.D. and Y.W. performed the parasitological evaluation of the compounds. T.D.O. performed the ultra-deep sequencing of resistant clones and the corresponding bioinformatics. R.R. and R.B. performed the PK studies and analysed the data. L.L. and V.M.A. investigated the gametocytocidal activity. A.M.-S., D.M. and H.J.V. investigated the activity of the compounds in mouse model systems. M.L. wrote the manuscript. All authors participated in discussion and manuscript editing.

## Additional information

**Competing financial interests:** The authors declare no competing financial interests.

