## [Peer Review File · Nature Communications]

Reviewers' comments:

Reviewer #1 (Remarks to the Author):

1. Line 258: 10.0 mg/ kg should read: "10.0 mg/kg/day"
2. On what basis did the authors not determine the activity of their lead compounds against liver stage parasites or other exoerythrocytic parasite stages?
3. The authors state that: "The compounds were inactive in the P. berghei rodent malaria model system." How were they administered and do they have an explanation for the lack of efficacy in P. berghei infected mice?
4. In Figure 5, some of the proposed metabolic transformations are unusual such as S-dealkylation of a sulfonamide functional group. The authors should include only the experimentally observed metabolic transformations.

Reviewer #2 (Remarks to the Author):

The manuscript entitled "SC83288, a novel clinical development candidate for the treatment of severe malaria" deals with a medicinal chemistry program which uses amicarbalide as a precursor. The authors carried out various chemical modifications of the starting compound and identified two molecules, SC81458 and SC83288, that can cure a P. falciparum infection in a humanized NOD/SCID gamma c mouse model system. Further the authors performed pre-clinical pharmacokinetic and toxicological studies and did ultra-deep sequencing of resistant parasites. By doing this, they identified PfATP6, a sarco/endoplasmic reticulum Ca²⁺ transporter, as a potential target. The manuscript is well written and all experiments are carried out carefully. The manuscript is of interest to the readership of NCOMMS, however prior considering the manuscript for publication, a revision is required.

- 1) The authors should detail the pharmacokinetical difference of the known starting compound amicarbalide and their two new compounds, since all reveal a similar IC-value.
- 2) The authors carried out animal experiments in humanized NSG mice, however they should consider to perform these experiments also in the P. berghei mouse model, since this might additional information about the druggability of the their compounds against different types of Plasmodium.
- 3) The author stated that SC83288 was stable in the presence of cultured human red blood cells and in P. falciparum cultures for at least 6 h, what was the stability of SC81458, since the molecules are quite "similar".
- 4) The authors crated drug resistant parasites and identified after ultra-deep sequencing mutations in Pf3D7_1447900 encoding PfMDR2, Pf3D7_0106300 encoding the Ca²⁺

transporting PfATP6 and Pf3D7_1241800 encoding a putative ATP-dependent RNA helicase DBP9. They were analysing PfATP6 more deeply by confocal microscopy using the Ca²⁺ sensitive fluorochrome Fluo-4. However the drugs did not affect the cytoplasmic free Ca²⁺ level. Since it would be highly beneficial to gain information about the mechanism of action the author should consider to perform pull-down experiments with their compounds to identify the target in *P. falciparum*.

Minor points:

Line 10 "44" should be "4"?

Line 361 "check" should be in past tense

Reviewer #3 (Remarks to the Author):

The paper by Pegoraro describes the identification of SC83288 as a proposed treatment for severe malaria. The compounds identified have single digit nanomolar activity in vitro versus cultured *Plasmodium falciparum* parasites and also have in vivo activity in a NOD/SCID mouse model when administered through ip injection. PK studies are performed in several animal models and PK/PD assessments are performed. The mechanism of action is unclear based on the studies performed.

The study is original and this is a new chemotype though I do wonder if there is any overlap with the mechanism of action of these analogues and that of diamidines such as DB75 (Purefield et al) (nucleic acid synthesis is targeted by DB75 at the trophozoite stage of development; DOI: 10.1186/1475-2875-8-104). The presentation and data is of good quality throughout but there are a number of conclusions, based on the data, that preclude it for being acceptable for publication.

It is not clear at all what is driving the medicinal chemistry optimization. Whilst Figure 2 provides a clear view of SAR it is not apparent how these changes are linked to the key profile ie. What is the target candidate profile (TCP) that was set and how do properties of the two potential candidates identified map onto to this target profile?

My own view is that SC83288 does not match against any of the priority TPPs for malaria (Medicines for Malaria Venture, MMV) because the oral bioavailability is so poor it can't be used orally. The impact of this is that this group has repositioned the series for IV use but the PRR data (Figure 4B, not done exactly as standard) suggests that it is a slower killer than artemisinins and the proposed candidate doesn't kill the ring stages - since speed of kill is essential in severe disease as shown by the Aquamat trial (Arts Vs Quinine, Lancet, 2010, 376, 1647) this drug will be inferior to ARTs - so no added value in this disease setting.

In Table 2 it is not clear what the final row applies to F(%)- this needs to be made clearer. One of the main issues with preclinical evaluation is the apparent lack of cellular toxicity

because the molecules likely do not penetrate membranes. PK/PD assessments are very well performed in this paper but what about NOEL and NOAEL values in the preclinical evaluation? What is the therapeutic window for an efficacious dose versus the no effect level. There is also no toxicokinetic evaluation of the key analogues.

The mechanism of action studies are not definitive - knock-out studies are required, or molecular studies introducing mutations to prove target -this is essential for publication in Nature Communications. As it stands the MOA aspect of the work does not strengthen the paper.

Overall, the argument for selection of this molecule as a candidate is limited by lack of knowledge of the mechanism of action, insufficient preclinical toxicity evaluation and a speed of kill that appears to be less than that of artesunate (Figure 4B).

Reviewer: 1

Comment 1: Line 258: 10.0 mg/ kg should read: "10.0 mg/kg/day"

Reply: The requested corrections have been made and consistency ensured (page 13, middle of first paragraph).

Comment 2: On what basis did the authors not determine the activity of their lead compounds against liver stage parasites or other exoerythrocytic parasite stages?

Reply: At the time we didn't think it was necessary to determine the activity of our compounds against extraerythrocytic stages since our compounds are intended for the treatment of severe malaria. However, given the reviewer's query we will gladly conduct these studies in the near future.

Comment 3: The authors state that: "The compounds were inactive in the P. berghei rodent malaria model system." How were they administered and do they have an explanation for the lack of efficacy in P. berghei infected mice?

Reply: The reviewer asks an interesting question. We have tested several of our compounds in the *P. berghei* and *P. vinckei* rodent malaria model systems. However, none of our compounds were efficacious against *P. berghei* and only a few including our two lead compounds revealed a therapeutic efficacy in *P. vinckei*. Differential therapeutic efficacy in various rodent model systems is a well-established phenomenon and is typically associated with charged compounds, such as positively charged amidines and negatively charged phosphoric acid or carboxylic acid containing drugs (references 27 to 29 in the manuscript). The phenomenon is explained based on selective uptake of charged compounds through parasite-induced channels, termed new permeation pathways (NPPs), in the host cell plasma membrane and species-specific characteristics of the NPPs (references 27 and 28 in the manuscript). We now include this explanation in the manuscript (page 13 last paragraph and following) and present an additional supplementary table compiling the results on the drug efficacy studies in rodent malaria model systems (Supplementary Table 8).

Comment 4. In Figure 5, some of the proposed metabolic transformations are unusual such as S-dealkylation of a sulfonamide functional group. The authors should include only the experimentally observed metabolic transformations.

Reply: We have revised Figure 5 as suggested by the reviewer and now show only experimentally observed metabolic transformations.

Reviewer: 2

*The manuscript entitled "SC83288, a novel clinical development candidate for the treatment of severe malaria" deals with a medicinal chemistry program which uses amicarbalide as a precursor. The authors carried out various chemical modifications of the starting compound and identified two molecules, SC81458 and SC83288, that can cure a *P. falciparum* infection in a humanized NOD/SCID gamma c mouse model system. Further the authors performed pre-clinical pharmacokinetic and toxicological studies and did ultra-deep sequencing of resistant parasites. By doing this, they identified PfATP6, a sarco/endoplasmic reticulum Ca²⁺ transporter, as a potential target. The manuscript is well written and all experiments are carried out carefully. The manuscript is of interest to the readership of NCOMMS, however prior considering the manuscript for publication, a revision is required.*

Comment 1: The authors should detail the pharmacokinetic difference of the known starting compound amicarbalide and their two new compounds, since all reveal a similar IC-value.

Reply: As mentioned in the manuscript (page 6, second paragraph) amicarbalide has been discontinued as a veterinary drug against babesiosis and related protozoan diseases in livestock and companion animals. The reasons given include slow parasite clearance rates, high relapse frequencies, undesirable mutagenic and toxic side effects, and manufacturing safety issues (references 21 and 22 in the manuscript). Since it lacks clinical application itself, we did not investigate amicarbalide any further and did not conduct any pharmacokinetic studies using this compound. Irrespective of this, we tried to obtain information regarding the PK of amicarbalide and searched the literature and contacted several companies. Unfortunately we could not obtain the requested information. Amicarbalide was discovered by May and Baker in 1960 (reference 20 in the manuscript). After several rounds of merger and acquisition May and Baker is now part of Rhone Poulenc, and they have no more records from that time. However, we see the reviewers point and now present PK data on the parental hit compound SC09064 and the close amicarbalide relative imidocarb (additional supplementary Figure 9). The data show that the pharmacokinetic properties of SC83288 are distinct from those of SC09064 and imidocarb with regard to AUC, plasma half-life, clearance, and volume distribution. These findings are described in the results section on page 15, end of second paragraph).

*Comment 2: The authors carried out animal experiments in humanized NSG mice, however they should consider to perform these experiments also in the *P. berghei* mouse model, since this might additional information about the druggability of the their compounds against different types of Plasmodium.*

Reply: We thank the reviewer for this suggestion. As also mentioned in our reply to comment 3 made by reviewer 1 we have tested several of our compounds in different rodent malaria model systems, including *P. berghei*. However, none of the compounds revealed any activity. In comparison, some of the compounds were active in the *P. vinckei* rodent system. Differential therapeutic efficacy in various rodent model systems is a well-established phenomenon and is explained based on selective uptake of charged compounds through parasite-induced channels, termed new permeation pathways (NPPs), in the host cell plasma membrane and species-specific characteristics of the NPPs (references 27 and 28 in the manuscript). We now include this explanation in the manuscript (page 13 second paragraph and following) and present an additional supplementary table compiling the results on the drug efficacy studies in rodent malaria model systems (supplementary Table 8).

*Comment 3: The author stated that SC83288 was stable in the presence of cultured human red blood cells and in *P. falciparum* cultures for at least 6 h, what was the stability of SC81458, since the molecules are quite "similar".*

Reply: As suggested by the reviewer, we now show data on the stability of SC81458 (additional supplementary Figures 7E to H). The results are described on page 12, third paragraph).

Comment 4: The authors crated drug resistant parasites and identified after ultra-deep sequencing mutations in Pf3D7_1447900 encoding PfMDR2, Pf3D7_0106300 encoding the Ca²⁺ transporting PfATP6 and Pf3D7_1241800 encoding a putative ATP-dependent RNA helicase DBP9. They were analysing PfATP6 more deeply by confocal microscopy using the Ca²⁺ sensitive fluorochrome Fluo-4. However the drugs did not affect the cytoplasmic free Ca² level. Since it would be highly beneficial to gain information about the mechanism of action the author should consider to perform pull-down experiments with their compounds to identify the target in P. falciparum.

Reply: We hoped that the SC resistant *P. falciparum* strains would reveal information regarding the mode of action of the compounds. Of the four candidate genes associated with SC resistance (see Figure 8) we further investigated PfATP6. However, we found no evidence of the SC compounds inhibiting the Ca²⁺ transporting activity of PfATP6, suggesting that PfATP6 is not a molecular target (see supplementary Figure 12). Studies evaluating the other candidate genes are ongoing in our laboratories. However, the data are too preliminary to be published at this stage. Since a full characterization of a drug's mode of action can be a laborious and time-consuming effort we kindly ask the reviewer to share our opinion that further investigation into the compounds mode of action is beyond the scope of this study.

Comment 5: Line 10 "44" should be "4"?

Reply: Done as suggested.

Comment 6: Line 361 "check" should be in past tense

Reply: Done as suggested.

Reviewer: 3

The paper by Pegoraro describes the identification of SC83288 as a proposed treatment for severe malaria. The compounds identified have single digit nanomolar activity in vitro versus cultured Plasmodium falciparum parasites and also have in vivo activity in a NOD/SCID mouse model when administered through ip injection. PK studies are performed in several animal models and PK/PD assessments are performed. The mechanism of action is unclear based on the studies performed.

Comment 1: The study is original and this is a new chemotype though I do wonder if there is any overlap with the mechanism of action of these analogues and that of diamidines such as DB75 (Purefield et al) (nucleic acid synthesis is targeted by DB75 at the trophozoite stage of development; DOI: 10.1186/1475-2875-8-104). The presentation and data is of good quality throughout but there are a number of conclusions, based on the data, that preclude if for being acceptable for publication.

Reply: Given the little structural overlap between our SC compounds and DB75 or its pro-drug DB289 (pafuramidine), as also pointed out by the reviewer, we think it is unlikely that our two development candidates and DB75 share the same mode of action, although we cannot exclude it at present since the mode of action of the SC compounds is still unclear.

Comment 2: It is not clear at all what is driving the medicinal chemistry optimization. Whilst Figure 2 provides a clear view of SAR it is not apparent how these changes are linked to the key profile ie. What is the target candidate profile (TCP) that was set and how do properties of the two potential candidates identified map onto to this target profile?

Reply: Criteria driving the campaign were *in vitro* antiplasmodial activity, solubility in water, membrane permeability, therapeutic efficacy in a rodent malaria model and eventually in the humanized NSG mouse model system, and an acceptable *in vitro* safety profile. We now clearly state this in the results section (page 8, first paragraph) and in the discussion (page 19, second paragraph). To clarify this issue we have now added two additional supplementary Tables providing primary information regarding water solubility and membrane permeability (supplementary Table 2) and therapeutic efficacy in rodent malaria model systems (supplementary Table 8).

Comment 3: My own view is that SC83288 does not match against any of the priority TPPs for malaria (Medicines for Malaria Venture, MMV) because the oral bioavailability is so poor it can't be used orally. The impact of this is that this group has repositioned the series for IV use but the PRR data (Figure 4B, not done exactly as standard) suggests that it is a slower killer than artemisinins and the proposed candidate doesn't kill the ring stages - since speed of kill is essential in severe disease as shown by the Aquamat trial (Arts Vs Quinine, Lancet, 2010, 376, 1647) this drug will be inferior to ARTs - so no added value in this disease setting.

Reply: We respectfully disagree with the reviewer on this point. SC83288 fulfills several criteria for antimalarial candidate drugs as specified by MMV (reference 41 in the manuscript). This includes IC₅₀ and IC₉₉ values <10 nM and ≤50 nM, respectively (Figure 3 and Table 1); killing ≥ 99.9% of the parasite population within the first 48 h of treatment (Figure 4 and Table 1); *in vitro* activity against drug resistant *P. falciparum* strains (supplementary Table 3); therapeutic efficacy in the humanized NSG mouse model system at a dose of < 10 mg kg⁻¹ body weight day⁻¹ (Figure 6); and low costs of goods.

The speed of action of SC83288 was evaluated in two different assays. In the assay first described by Vial et al. (2004) (reference 7 in the manuscript), SC83288 inhibited parasite growth significantly faster than did artemisinin (Figure 4A). In the second assay, which, in addition to growth inhibition, also takes recrudescence into consideration (assay developed by Sanz et al. (2012) (reference 26 in the manuscript), SC83288 was indeed slightly inferior to artemisinin (Figure 4B). However, recrudescence is not a major concern when treating severe malaria, whereas a rapid block of parasite growth is. In this regard, SC83288 is as good as or even better than artemisinin. Our finding that SC83288 reveals no cross resistance with artemisinin represents an important added value of SC83288, particularly, in light of the spreading resistance to artemisinin and its derivatives and given the current paucity of alternatives for the treatment of severe malaria. We do not wish to over promote SC83288 and we hope we have provided a balanced assessment of the compound's profile.

Comment 4: In Table 2 it is not clear what the final row applies to F(%)- this needs to be made clearer. One of the main issues with preclinical evaluation is the apparent lack of cellular toxicity because the molecules likely do not penetrate membranes. PK/PD assessments are very well performed in this paper but what about NOEL and NOAEL values in the preclinical evaluation? What is the therapeutic window for an efficacious dose versus the no effect level. There is also no toxic kinetic evaluation of the key analogues.

Reply: We thank the reviewer for pointing out to us that we neglected to mention the NOEL and NOAEL of the compound. We now state these parameter in the revised manuscript:

“The maximal tolerated i.v. bolus dose (i.v. MTD) of SC83288 was at 30 mg kg⁻¹ body weight (extrapolated C₀ of 65,600 ng ml⁻¹) and 15 mg kg⁻¹ body weight (extrapolated C₀ of 27,300 ng ml⁻¹) in rats and mice, respectively. Observed adverse effects included ataxia, respiratory symptoms, and sleepiness. However, all treated animals fully recovered within 30 min after drug application. Hematological and clinical biochemical parameters were assessed 24 hrs post application and remained within normal range in all animals (Supplementary Table 7). Increasing the dose to 45 mg kg⁻¹ body weight caused apnea in rats. The no-observed-adverse-effect level (NOAEL) and the no-observed-effect level (NOEL) corresponded to a tested i.v. bolus dose of 22.5 mg kg⁻¹ body weight (extrapolated C₀ of 49,200 ng ml⁻¹) and 15 mg kg⁻¹ body weight (extrapolated C₀ of 32,800 ng ml⁻¹) in rats, respectively.” (page 11, end of second paragraph).

It is correct that the toxicokinetics of SC83288 is currently unclear. We fully acknowledge this in the manuscript (page 19, end of first paragraph). Pre-regulatory toxicokinetic studies, including 28 day repeat-dose toxicology, are currently ongoing in dogs. Since these studies are complex and time-consuming, the results are not expected before spring of next year and are, therefore, beyond the scope of this study.

With regard to the therapeutic window of the compound, in supplementary Figure 10 we have simulated different dosing regimens in humans, based on an allometric scaling of the PK studies conducted in different animal species and on the therapeutic efficacy studies conducted in the humanized mouse challenge model. We found that an infusion rate between 8.8 mg h⁻¹ and 13.1 mg h⁻¹ administered over a period of less than 8 h suffices to reduce the initial parasite burden by 99.9 % within 48 h. This projected dosing regimen would maintain a safety margin of ~ 55-fold assuming comparable i.v. NOAEL in rats and humans with a C₀ of 49,200 ng ml⁻¹. We explain this in the discussion (page 22).

F(%) is now explained in the legend to Table 2 and included in the list of abbreviations.

Comment 5: The mechanism of action studies are not definitive - knock-out studies are required, or molecular studies introducing mutations to prove target -this is essential for publication in Nature Communications. As is stands the MOA aspect of the work does not strengthen the paper.

Reply: We fully agree with the reviewer that the mode of action of SC8322 is currently unclear. We hoped that the SC resistant *P. falciparum* strains would reveal information regarding the mode of action of the compounds. Of the four candidate genes associated with SC resistance (see Figure 8) we further investigated PfATP6. However, we found no evidence of the SC compounds inhibiting the Ca²⁺ transporting activity of PfATP6 (supplementary Figure 12), suggesting that PfATP6 is not a molecular target. Studies evaluating the other candidate genes are ongoing in our laboratories. However, the data are too preliminary to be published at this stage. Since a full characterization of a drug's mode of action can be a laborious and time-consuming effort we kindly ask the reviewer to share our opinion that further investigation into the compounds mode of action is beyond the scope of this study.

Reviewers' comments:

Reviewer #1 (Remarks to the Author):

NONE

Reviewer #2 (Remarks to the Author):

The authors commented satisfactory to the suggestions of this reviewer and changed the manuscript accordingly. Therefore the manuscript is suitable for acceptance.

Reviewer #3 (Remarks to the Author):

Whilst the majority of points raised have been adequately dealt with the key issue raised in comment 3 still applies in that delivery of an oral molecule is the key component of MMVs drug discovery TCPS. There is no role for an IV antimalarial for severe disease with a slower PRR (parasite reduction ratio) than the artemisinins as it could never demonstrate superiority clinically. The molecule described here clearly acts on late trophozoites, confirmed by the authors in the text, and the current understanding of the fundamentals of parasite kill would predict a PRR of such a compound to be similar to the 4-aminoquinolines and significantly slower than the artemisinins as it can only exert an effect on a limited proportion of the parasites asexual life cycle (mid to late trophozoite).

I believe the interpretation presented by the authors is misleading and incorrect. The methods and data they refer to in the text and rebuttal are non-standard approaches prone to give erroneous information specifically with respect to kill dynamics. This is the reason MMV adopted PRR as a discriminator in their discovery programmes. I think as a very minimum requirement, and before consideration for publication, the authors should be asked to arrange for an independent laboratory to perform the PRR assay as defined by MMV in order to confirm the actual PRR for SC83288 .This is a key fundamental criticism of the molecule, the paper and the proposed use of the molecule. Failure to definitively demonstrate this characteristic would negate any interest in this compound as it has no value as an oral treatment due to very poor bioavailability noted above.

Reviewer: 1

NONE

Reviewer: 2

The authors commented satisfactory to the suggestions of this reviewer and changed the manuscript accordingly. Therefore the manuscript is suitable for acceptance.

Reviewer: 3

Comment: Whilst the majority of points raised have been adequately dealt with the key issue raised in comment 3 still applies in that delivery of an oral molecule is the key component of MMVs drug discovery TCPS. There is no role for an IV antimalarial for severe disease with a slower PRR (parasite reduction ratio) than the artemisinins as it could never demonstrate superiority clinically. The molecule described here clearly acts on late trophozoites, confirmed by the authors in the text, and the current understanding of the fundamentals of parasite kill would predict a PRR of such a compound to be similar to the 4-aminoquinolines and significantly slower than the artemisinins as it can only exert an effect on a limited proportion of the parasites asexual life cycle (mid to late trophozoite).

I believe the interpretation presented by the authors is misleading and incorrect. The methods and data they refer to in the text and rebuttal are non-standard approaches prone to give erroneous information specifically with respect to kill dynamics. This is the reason MMV adopted PRR as a discriminator in their discovery programmes. I think as a very minimum requirement, and before consideration for publication, the authors should be asked to arrange for an independent laboratory to perform the PRR assay as defined by MMV in order to confirm the actual PRR for SC83288. This is a key fundamental criticism of the molecule, the paper and the proposed use of the molecule. Failure to definitively demonstrate this characteristic would negate any interest in this compound as it has no value as an oral treatment due to very poor bioavailability noted above.

Reply: As I understand the reviewer's comments, he/she requests confirmation of the parasite reduction ratio (PRR) described in our manuscript for the compound SC83288 (Figure 4b) by an independent laboratory since he/she is of the opinion that we used a non-standard (not MMV approved) assay to determine the parameter. First, the assay we used is MMV approved (see the following url on the MMV homepage: <http://www.mmv.org/newsroom/publications/p-falciparum-vitro-killing-rates-allow-discriminate-between-different>). In fact, the original publication of the assay by Sanz et al. (PLoS One. 2012;7(2):e30949) is co-authored by a member of the MMV team. So unless the reviewer can specify why this is a non-standard assay or how we have misunderstood the original protocol, I see no reason to question our findings. In response to the reviewer's comments we now clearly state in the manuscript that the *in vitro* PRR assay used in our study was performed according to MMV approved standards (see methods page 26).

Secondly, I am of the opinion that it is not a legitimate request by a reviewer to have data reproduced by an independent party. Of course only robust and fully reproducible data should be published. We fully subscribe to this cornerstone of good scientific practice. However, it is

primarily the responsibility of the authors to ensure that their data are reproducible and, moreover, validated using independent assays.

Still in line with this point, different research groups in our study with different approaches show highly consistent results. In the case of the *in vitro* PRR assay challenged by the reviewer, the experiment has been repeated four times in completely independent biological replicates (as stated in the manuscript) and the findings have been validated in a humanized *P. falciparum*-infected mouse model system (Figure 6 and supplementary Figure 8). Both the *in vitro* and the *in vivo* assays yielded comparable results, namely a logPRR value of 3.0 (Table 1). I would like to point out that the *in vitro* PRR assay was performed in my laboratory at Heidelberg University, whereas the *in vivo* PRR assay conducted in infected mice was performed in the laboratory of Alica Moreno-Sabater at the Centre d'Immunologie et des Maladies Infectieuses in Paris. Thus, two different assays and two different teams came to the same conclusion. With regard to the question of how to interpret a logPRR value of 3.0 I would like to refer to a strategy paper released by MMV where on page 7 it says "minimum criteria for any antimalarial: • A PRR of >10, ideally more than 10^3 kills per cycle (Strategy for the selection and early development of combination drugs for the treatment of uncomplicated *P. falciparum* malaria. 7th September 2010, MMV, Geneva, Switzerland). Thus, a logPRR of 3.0 (10^3) is considered by MMV an ideal value for a drug candidate. We have amended the text as follows to clearly state possible limitations of the compound in relation to artemisinin with regard to the killing rate:

"A logPRR of ≥ 3.0 is considered a favorable property of an antimalarial drug candidate ⁴¹, although the value is lower than that of artemisinin ²⁶." (discussion, first paragraph, page 19).

The reviewer further states that "there is no role for an IV antimalarial for severe disease with a slower PRR (parasite reduction ratio) than the artemisinins as it could never demonstrate superiority clinically." We respectfully disagree with the reviewer. If artemisinin resistance continues to spread, and taking into consideration that a substantial number of patients treated with parenteral artesunate develop severe adverse side effects, then there is clearly a role for an alternative IV treatment for severe malaria, even though the compound may not match artemisinin's pharmacokinetic and pharmacodynamic properties in all aspects. To clarify this issue we have expanded the discussion by the following sentences:

"In summary, SC83288 may not match the favorable pharmacological and pharmacodynamics properties of artemisinin in all aspects. However, the attributes of SC83288 have to be considered in light of the emerging resistance against artemisinin derivatives and the adverse side effects observed in a substantial number of patients treated with parenteral artesunate for severe malaria ^{3, 4, 16, 17}. The role of SC83288 as an alternative chemotherapeutic option will require extensive clinical studies. However, the available preclinical data suggest that SC83288 holds promise for an efficacious and safe clinical candidate for the acute treatment of severe malaria." (end of discussion on page 22).